# Weakening and warming of the European Slope Current since the late 1990s attributed to basin-scale density changes.

Matthew Clark[1], Robert Marsh[1], James Harle[2]

[1] Ocean and Earth Science, University of Southampton Waterfront Campus, National Oceanography Centre, European Way, Southampton, SO14 3ZH, UK.
[2] National Oceanography Centre, European Way, Southampton, SO14 3ZH, UK

*Correspondence to*: Matthew Clark (matt.clark@soton.ac.uk)

**Abstract.** Oceanic influences on shelf seas are mediated by flow along and across continental slopes, with consequences for regional hydrography and ecosystems. Here we present evidence for the variable North Atlantic influence on European shelf seas over the last four decades, using ocean analysis and reanalysis products, and an eddy-resolving ocean model hindcast. To first order, flows oriented along isobaths at the continental slope are related to the poleward increase of density in the adjacent deep ocean that supports a geostrophic inflow towards the slope. In the North Atlantic, this density gradient and associated inflow has undergone substantial, sometimes abrupt, changes in recent decades. Inflow in the range 10-15 Sv is identified with eastward transport in temperature classes at 30°W, in the latitude range 45-60°N. Associated with major subpolar warming around 1997, a cool and fresh branch of the Atlantic inflow was substantially reduced, while a warm and more saline inflow branch strengthened, with respective changes of the order 5 Sv. Total inflow fell from ~15 Sv pre-1997 to ~10 Sv post-1997. In the model hindcast, particle tracking is used to trace the origins of poleward flows along the continental slope to the west of Ireland and Scotland, before and after 1997. Backtracking particles up to 4 years, a range of subtropical and subpolar pathways is identified from a statistical perspective. In broad terms, cold, fresh waters of subpolar provenance were replaced by warm, saline waters, of subtropical provenance. These changes have major implications for the downstream shelf regions that are strongly influenced by Atlantic inflow, the northern North Sea in particular, where "subtropicalization" of ecosystems has already been observed since the late 1990s.

## 1 Introduction

Over recent decades, the European shelf seas have undergone profound changes in hydrography, biogeochemistry and ecosystems. Major ecosystem changes are most notable in the North Sea. Continuous Plankton Recorder (CPR) data indicate that North Sea ecosystems underwent a regime shift over 1982-88, from a "cold dynamic equilibrium" of 1962-83 to a "warm dynamic equilibrium" of 1984-99 (Beaugrand, 2004). At species level, impacts have been profound: northward shifts in copepod populations (Beaugrand, 2004); steady increase in squid catch since 1980 (van der Kooij et al., 2016); "subtropicalization" of pelagic fish communities (Montero‐Serra et al., 2015); evidence that cod recruitment and herring spawning stocks are highly responsive to temperature in the northern North Sea (Akimova et al., 2016).

While these poleward range shifts have been directly attributed to local warming (Beaugrand, 2004), a natural conduit for range expansion is the European Slope Current (henceforth, 'Slope Current'), conveying warm-affinity species from equatorward latitudes to the North Sea. Furthermore, Slope Current water and flow is highly variable. Observations at the "Extended Ellett Line" (EEL) repeat hydrographic section off western Scotland indicate background warming and declining nutrient concentrations in upstream flows, from 1996 to the mid-2000s (Johnson et al., 2013). Meanwhile, peak inflows of warm, salty Atlantic Water coincide with Chlorophyll-*a* increases, despite declines in nutrient concentration (McQuatters-Gollop et al., 2007). CPR data indicate that periodic changes in zooplankton in the North Sea are directly related to variations of Atlantic inflow (Reid et al., 2001;Reid et al., 2003). In conclusion, local warming alone may not explain the

rapid, sometimes abrupt, and ongoing changes in community structure and ecosystem functioning in the North Sea. It is likely that the Slope Current provides the connectivity necessary for bottom-up (food supply) and/or top-down (predator) drivers of change.

    Variability of the Slope Current related to ecosystem change (Reid et al., 2001;Reid et al., 2003) may be related to basin-

scale events (Marsh et al., 2017). The North Atlantic is a highly dynamic ocean basin, dominated by two gyre systems: the Subpolar Gyre (SPG) and the Subtropical Gyre (STG). The Gulf Stream flows between these two and eventually becomes the North Atlantic Current (NAC), also referred to as the North Atlantic Drift. The NAC brings warmer Atlantic waters from the subtropics across to the eastern boundary and continues north-eastwards: taking a multi-route path including branches flowing through the Rockall Trough (Holliday et al., 2018). The NAC eventually flows beyond the UK and into the

Norwegian Sea. Here, the water cools and sinks, forming the cold, deep return flow which completes the North Atlantic Overturning Circulation (AMOC). However, not all of the water follows this pathway. Some of the NAC, which has multiple branches, is entrained into the European Slope Current which flows around the European Shelf (Marsh et al., 2017). Flowing northward along the shelf break and through the Rockall Trough, the current at times is deflected onto the shelf (Porter et al., 2018). However, the majority continues past the north of Scotland. Some of this water follows the path of the Norwegian

Current but some follows the northeast Scottish coastline, onto the shelf and into the North Sea. Some of the Slope Current water that travels further north is eventually deflected into the northern North Sea via East Shetland Inflow and Norwegian Trench inflow (Holliday and Reid, 2001) Previous estimates suggest that up to 40% of Slope Current water ends up in the North Sea (Marsh et al., 2017), thus a major input to the European shelf seas.

Gradients of sea surface height are also a factor in toward-shelf and along-shelf transport. Empirical orthogonal function analysis of sea surface height variability across the North Atlantic subpolar gyre has previously been used as a proxy for a "subpolar gyre index" (Hátún et al., 2005;Hátún and Chafik, 2018). Changes to the subpolar gyre index correlate strongly with pulses of salinity observed in the northern North Sea (Pätsch et al., 2020). Satellite altimetry data has been used to produce a gyre index, which is significantly correlated with the North Atlantic Oscillation (NAO) index (Pingree, 2002).

Extreme low winter NAO index conditions (weaker than usual air pressure differences between Iceland and the Azores) are found to enhance poleward flow along the European shelf edge (Pingree, 2002). The NAO has also been shown to affect surface temperatures along the eastern boundary: strong negative NAO in the months leading to January has been shown to enhance shelf-edge winter warming and current strength along the Iberian Poleward Current, aka "Navidad years" (Garcia-Soto et al., 2002).


    The Slope Current is a narrow bathymetry-constrained, northward-flowing shelf edge current. It is primarily driven by geostrophic inflow in proportion to the poleward density gradient in the North Atlantic basin (Marsh et al., 2017). It is therefore likely that changes to the density distribution of the North Atlantic could have profound effects on the Slope Current transport and composition, and the wider shelf edge and shelf sea environment. The Slope Current is also partially

driven by onshore Ekman flow associated with poleward winds parallel to the coastline and shelf edge (Xu et al., 2015). Associated vertical mixing and flow instabilities at the shelf break provide an important mechanism for drawing essential nutrients to the shelf seas (Holt et al., 2009;Huthnance et al., 2009;Mathis et al., 2019). The Slope Current is often characterised in literature as having a high salinity and velocity core at approximately 200 – 300 m above the 600 m bathymetry contour (Porter et al., 2018). The subpolar North Atlantic underwent extensive warming in the second half of the

1990s (Marsh et al. 2008), which was sustained well into the 2010s. Superimposed on this warming has been more recent short-term (interannual) variability: temperatures of up to 2 °C less than the mean were observed in the SPG from 2013, peaking in 2015 (Duchez et al., 2016;Josey et al., 2018). Coupled with this cooling was a sharp and record-breaking decline

in salinity of the subpolar North Atlantic and shelf edge after 2015, associated with a strong positive NAO index over the SPG (Holliday et al., 2020). The NAC is influential in conveying material such as pollutants, nutrients and organisms including their larvae into changing shelf sea environments (Huthnance et al., 2009;Reid et al., 2003), which are becoming more or less conducive to any given species, and potentially more conducive to invasive species (Holt et al., 2018;Porter et al., 2018). The SPG and NAC are also known to have significant effects on the geographical shifts of plankton, fish, and whales in the ocean basin and shelf seas. Stronger phases of the SPG have been shown to reduce the amount of plankton and fish observed at the shelf edge and in the Rockall-Hatton Plateau (Hátún et al., 2009). Changes to certain seabird populations such as kittiwakes, terns and auks have been strongly correlated to the SPG index: more specifically, high breeding success coincided with an expanded SPG, increasing zooplankton availability for the fish they feed on (Hátún et al., 2017). Conversely, a reduction of the AMOC and contraction of the SPG has previously been suggested in coupled climate-ecosystem models as a mechanism for a suppressed ecosystem in the North Atlantic (Schmittner, 2005). In summary, the Atlantic influence on European shelf seas is spatially and temporally variable. The strongest and most variable influence of Atlantic Water is felt along the shelf break and in the northern North Sea (Koul et al., 2019). The Slope Current is a major pathway for bringing waters of sub-polar and sub-tropical origin to the shelf edge and shelf seas. Warmer water arriving from the North Atlantic into the European shelf seas has already been attributed to changes in the ecology and nutrients of UK waters (Holliday and Reid, 2001;Stebbing et al., 2002;Reid et al., 2003;Mathis et al., 2019). Given these changes, it is therefore timely to determine the effects of the changing North Atlantic on Slope Current provenance.

This study has been split into three main aims: (1) to identify the geostrophic inflow to the Slope Current; (2) to examine and where possible quantify the effects of changes of this inflow on the Slope Current; (3) to assess and quantify the changing provenance of the Slope Current waters. The rest of the paper is outlined as follows. In Sect. 2, we outline datasets and diagnostics. In Sect. 3, we first examine sub-surface hydrographic variability across the mid-latitudes of the North Atlantic along the eastern shelf break over 1980-2019, followed by analysis of zonal transport variability across mid-latitudes; we then focus on shelf edge transport and establish Lagrangian evidence for the changing provenance of Slope Current water before and after 1997. In Sect. 4, we synthesize these findings in the context of previous studies of large-scale and regional change in the northeast Atlantic and shelf seas.

## 2 Datasets and Methods

We use several data sources and resources: an ocean reanalysis product and an eddy-resolving global ocean model hindcast. Lagrangian computational particle tracking experiments have been conducted to determine the provenance of shelf edge and Slope Current waters. These datasets and diagnostics are outlined in the following sub-sections. The code used for the data analysis mentioned below and plotting is available separately via Zenodo (Clark et al., 2021).

### 2.1 Gridded reanalysis datasets

For a consistent physical and dynamical description of the North Atlantic, the NOAA gridded dataset known as "GODAS: Global Ocean Data Assimilation System" (NOAA, 2019;Nishida et al., 2011) has been used here. The dataset provides monthly values of temperature, salinity, and zonal/meridional components of velocity for 1980 – 2019 inclusive. The dataset assimilates measurements from a suite of observational tools, including XBTs, Argo floats, moorings and satellite altimeters (Nishida et al., 2011). Resolution for all parameters is 1/3° latitude by 1° longitude, at 40 depth levels. GODAS data is an assimilation of multiple measurement techniques and data sources. We acknowledge that, because of the small scale of the Slope Current, the current itself is not resolved in this dataset. However, it does provide data in sufficient resolution to assess the state and changes in the North Atlantic basin, particularly the changes within the subpolar gyre. As an assimilated and

gridded dataset, GODAS uses a 3DVAR assimilation scheme to plug gaps in the data, especially at depth (NOAA, 2019). GODAS salinity is mostly "synthetic": it uses a computed salinity profile using the annual T-S relationship of a region (NOAA, 2019;Behringer and Xue, 2004). Despite the fact that GODAS "seriously underestimates salinity variability" (NOAA, 2019), since the density and hence the geostrophic velocity is mainly influenced by temperature variability in the North Atlantic region (based on the analysis presented in the results), it is appropriate for looking at longer term (decadal scale) North Atlantic variability, where density anomalies are predominantly associated with temperature variability. Eastward and northward velocity field is also included in the analysis. For this work, we created a climatology of mean temperature, salinity and density at each chosen depth level, for the entire time series.

Monthly climatologies of temperature and salinity were constructed for the entire time series by taking the mean average temperature and salinity per month. Using the calculated climatologies, we have produced anomaly maps (for any selected depth) and meridional profiles, in the form of Hovmöller diagrams, along the shelf edge to capture the greatest density gradient of temperature and salinity. The shelf edge was defined as the first grid cell, reading from east to west, containing data at a grid depth of at least 600 m.

## 2.2 Transport calculations and metrics

Using the Python "Gibbs Sea Water" package, hosting the full Equation of State of Seawater TEOS10 (IOC et al., 2010), density was calculated from the temperature and salinity. The density was then used to calculate the geostrophic eastward volume transport of water through sections, using the (eastward) Thermal Wind Equation (Equation 1):

$$\frac{\partial u}{\partial z} = \frac{g}{\rho_0 f}\frac{\partial \rho}{\partial y}, \tag{1}$$

where $\frac{\partial u}{\partial z}$ = eastward velocity change over depth, $g$ = gravitational acceleration, $\rho_0$ = reference density, $f$ = Coriolis parameter, $\frac{\partial \rho}{\partial y}$ = change of density over latitude.

Integrating within the upper 1000m of water between 45 – 60 °N to obtain geostrophic volume transport ($U_{geo}$) through the section:

$$U_{geo} = \int_{-1000\,m}^{0\,m} \int_{45\,°N}^{60\,°N} \frac{\partial u}{\partial z}\,dz = \left(\left(\sum (u \times \Delta z)\right) \times \Delta y\right)/1 \times 10^6 \tag{2}$$

where $\Delta z$ is the depth step, $\Delta y$ is the change in latitude, and dividing by $1 \times 10^6$ provides the transport in Sverdrups (Sv, where Sv = $1 \times 10^6$ m$^3$s$^{-1}$).

We further use the eastward component of currents (part of the absolute velocity) in the GODAS dataset to calculate $U_{tot}$, the total transport in the upper 1000 m. We also calculate two other different transport types, to help determine the changing provenance of Slope Current waters: large-scale Sverdrup transport, calculated from NCEP wind field (gridded atmospheric wind on a 2.5° regular latitude/longitude grid, updated to 2020 (Kalnay et al., 1996)), which is used as a proxy of subpolar gyre circulation; and Ekman transport at the shelf edge (15 °W, 50 – 58 °N), based on the GODAS wind stresses. Please note that transport in our results and discussion is assumed to be volumetric water transport unless otherwise stated.

Sverdrup balance:

$$\beta V = \frac{1}{\rho}\left(\frac{\partial \tau_{sy}}{\partial x} - \frac{\partial \tau_{sx}}{\partial y}\right) = \frac{\mathrm{curl}\boldsymbol{\tau}_s}{\rho} \tag{3}$$

where $\beta$ is the meridional derivative of the Coriolis parameter (units $m^2\ s^{-1}$), V is meridional transport, $\rho$ is mean density (1025 kg $m^{-3}$), $\tau_{sy}$ and $\tau_{sx}$ are surface wind stress in the northward and eastward directions respectively. For a given latitude, we integrate V westward to obtain the total northward Sverdrup transport ($m^3\ s^{-1}$) and select the maximum value in the subpolar range of latitudes as an index of the wind-driven subpolar gyre.

Ekman volume transport (eastward):

$$U_{Ekman} = \sum\nolimits_{50\ ^\circ N}^{58\ ^\circ N}\left(\left(\frac{\tau_{ys}}{\rho f}\right)\Delta y\right)/1 \times 10^6 \tag{4}$$

where $\tau_{ys}$ is northward wind stress, $\rho$ is mean density (1025 kg $m^{-3}$) and f is the local Coriolis parameter.

We also calculate monthly SSH gradients (from GODAS sea surface height relative to the geoid (SSHG) data) along the same meridional transect (30 °W, 45 – 60 °N), to provide a proxy for SSH gradient associated eastward barotropic transport.

## 2.3 Eddy-resolving model output analysis

The GODAS assimilated values have been compared against and complimented with a 1/12° resolution eddy-resolving
global ocean model: ORCA12, part of the NEMO family of models (Madec, 2015). In previous studies, the model has been confirmed to be a good representation of North Atlantic subpolar circulation (Marzocchi et al., 2015). Quiver plots of velocity (Figure 1) have been produced for the ORCA12 250 m depth level to visually represent the currents at the shelf edge. This provided the evidence showing that shelf-edge flows, including the Slope Current, are resolved in this model. The northward velocity component of the model output has been integrated to produce an estimate of total northward volume
transport at the shelf edge. A similar calculation was used to calculate eastward transports in the subpolar North Atlantic. The transport calculation was limited to the upper 1000 m of the water column. This acts as an indicator for Slope Current transport.

## 2.4 Particle trajectory calculations

We use virtual particles to represent water parcels recruited to the Slope Current. To examine the origin of Slope Current waters, two particle tracking simulations were performed using ARIANE particle modelling software (Blanke and Raynaud, 1997), calculating trajectories backwards through time in the 'qualitative mode' of ARIANE. The ORCA12 model 5-day means (Madec, 2015) were used to provide the state of the ocean for the simulated years, in a sub-domain of the global ocean (constrained by computational demands), with a southern limit at 34.5°N and a northern limit ranging – due to the
tripolar mesh of ORCA12 – from 64°N (around 85°W) to 72.5°N (around 30°W). A number of particles were back-tracked in proportion to the northwards transport at the NW European shelf edge between 54.2 – 58.4 °N, a latitude range over which the Slope Current recruits Atlantic inflow; as such, the number of back-tracked particles varied with each experiment. Releases were designed to target the "core" of the Slope Current: ORCA12 gridcells in the upper 300 m were defined as a release location if there was northward transport present within 50 km of the 300 m isobath. The model co-ordinates of these
gridcells provide the initial positions for particles, which were distributed evenly across the vertical dimension of each gridcell, in proportion to the northward component of flow, with 1 particle allocated per 5 cm $s^{-1}$. As such, our 1992

ARIANE hindcast had 16470 releases, and our 2010 hindcast had 3626 releases, reflecting the reduction of northward transport within the stated latitude, longitude and depth limits.

Particles were released monthly, at the start of each month, for each chosen release year, and back-trajectories were calculated for a maximum of 4 years from the time of release; many particles reached the domain limit within 4 years, in which case their trajectories were terminated. The calculations provided daily particle location (latitude, longitude, and depth) as well as in-situ temperature, salinity and potential density (relative to the surface). Subsequent to these calculations, particle data are statistically analysed at selected on a grid of resolution 0.5° x 0.5° for areal fraction (percent cover of sea

surface) and particle mean age (days adrift, prior to arrival in the Slope Current), depth and potential temperature. We further record particles crossing 30 °W in the latitude range 40-60 °N, binned per 0.5° of latitude to obtain the corresponding percentages, along with mean age (days before arrival in the Slope Current), temperature and depth. Back-trajectories and associated statistics were thus calculated for the periods 1992 – 1988 and 2010 – 2006, addressing changes observed in the earlier dataset analysis, in particular the shift to a warmer subpolar North Atlantic.

**3 Results**

In the following section, we first examine sub-surface hydrographic variability in the northern subtropical and subpolar North Atlantic, and along the eastern shelf break, from the GODAS dataset. We then examine the variability of zonal transport across mid-latitudes. Finally, we examine shelf edge meridional transport and present Lagrangian evidence for changing provenance of the Slope Current over the last 40 years.

**3.1 Sub-surface hydrographic variability across mid-latitude North Atlantic and along the eastern shelf break over 1980-2019**

Since the Slope Current is defined as being a sub-surface feature with a core of 200 – 300m (Porter et al., 2018), we focus on changes to the mean decadal density (Figure 2) anomalies at the GODAS depth layer of 205 m, calculated from the temperature and salinity data using the equation of state of seawater (section 2.1). Other depth levels were tested (not shown)

but 205m gave the strongest gradients in density at a depth likely to influence the Slope Current. Salinity and temperature are closely related throughout the sub-surface North Atlantic Ocean. Where temperature is higher (lower) than the decadal mean, the water is typically more saline (fresher). This pattern is observed in all decades (not presented here). The trend in temperature anomaly is from a cool subpolar gyre to a warm subpolar gyre, although periods of intense (2 °C) cold temperature anomalies were observed in the subpolar region, particularly around 2011 - 15. Density shows an inverse

relationship with temperature and the patterns of variability match almost exactly. This is not always true when we compare the salinity to the density anomalies, especially nearer the Labrador Sea, where salinity anomalies are particularly large. Over the study period, there have been notable 'regime shifts' in temperature and density (and to a lesser extent, salinity) of the North Atlantic. The patterns of temperature and density anomalies have changed considerably over time, to the point at which the 2010s patterns are an approximate inverse of the 1980s patterns. Figure 2 compares the mean density anomaly

(with respect to the 1980 – 2020 climatology) pre and post-1997, to assess the impact of the warming of the subpolar North Atlantic on density patterns in the region. The subpolar region transitioned from a positive density anomaly of approximately 0.13 kg m$^3$ to a negative anomaly of approximately 0.11 kg m$^3$. The subtropical regions, especially just south of Cape Hatteras showed an even larger swing of density anomaly, exceeding ±0.18 kg m$^3$. These changes will have profoundly affected the density gradients. SSH anomalies (Figure 3) show large changes too. Where positive density anomalies are

present, there are negative SSH anomalies (and vice-versa), with an overall anomaly switch of ±0.11 m from pre to post 1997, in the subpolar gyre region: with anomalies of ±5 cm along 30 °W; in particular, the gradient of SSH (down-slope)

from the central northern subtropical gyre to the eastern subpolar gyre reduces by ~20 cm from pre-1997 to post-1997, primarily associated with subpolar warming and reduced densities.

When examining temperature and salinity as meridional profile time series along-slope, the same strong temperature-density relationship is observed. Hence, depth-averaged (0 to the nearest 600m depth bin) along-slope density anomaly for GODAS (Figure 4) have been plotted. The horizontal bands seen at 48 – 49 °N and 55 °N in Figure 4 are an artefact of jumping between grid cells to trace the shelf edge. From 1980 to 1987, the GODAS time series is dominated by a positive density anomaly. A short-lived switch to a negative anomaly is observed at 1990 before a more sustained positive density anomaly

to 1997. From 1997 onwards, the profile is dominated by negative anomalies, peaking just after the year 2000 at lower latitudes (< 48 °N). This negative density anomaly persists at most of the shelf edge until 2010. From 2010 onwards, anomalies vary between positive and negative values. The density anomalies in Figure 4b, dominated by temperature anomalies, are indicative of variable properties of water circulating in the Slope Current system, in turn related to changing provenance – a theme we return to in Sect. 3.3.


### 3.2 Zonal transport variability across North Atlantic mid-latitudes over 1980-2019

Geostrophic eastward volume transport (Figure 5a), calculated via the "thermal wind" equation as detailed in section 2.1, and absolute eastward volume transport (Figure 5b) has been calculated at 30 °W, between 45 – 60 °N. Transport was calculated monthly and then annual means taken to remove the large seasonal variability (not shown). A "temperature separation"

threshold of 11°C was determined as appropriate by testing different temperatures in the range 10 – 12 °C, based on the mean decadal temperature anomalies in the subpolar North Atlantic (subpolar gyre region, not shown).

From 1995, the transport begins to switch to a warmer regime (≥11 °C). By 1999, the cool geostrophic transport decreased to between 0 – 1 Sv, with some very brief periods of negative (westward) transport of magnitude < 1 Sv. In both the

geostrophic transport and the total transport (Figure 5), the cold pathway does begin to grow once again from 2012 in. However, total transport does not show any sign of increasing to pre-1997 levels.

The GODAS geostrophic component (Figure 5a) of transport strongly relates to the peaks shown in the total transport time series in Figure 5b. Total absolute velocity through the same section also shows the same general trends and peaks, but the

magnitude of the transports is approximately 5 to 6 Sv more. The total transport decrease after the 1995 – 97 regime shift is evident in both the geostrophic and total eastward transport estimates, with an observed decrease of approximately 10 Sv. The geostrophic transport estimates equate to the baroclinic, temperature and density-driven part of transport, which is slowly changing in relation to the changing density distribution as shown in Figure 2 and Figure 4. The total transport therefore includes additional contributions to transport below 1000 m, including any deep flows that may be identified with

barotropic transport.

To provide further metrics of the regional and boundary circulation, we calculate the basin-scale Sverdrup transport (Figure 6a), wind-driven Ekman eastward transport in the vicinity of the Slope Current (Figure 6b), and a mid-basin sea-level slope anomaly as an index of geostrophic surface flow (Figure 6c). All three indices reveal a range of monthly and interannual

variability. The annual amplitude of Sverdrup transport is on the order of 60 Sv, although annual-mean transport in the approximate range 25-35 Sv is much more stable. Ekman transport can vary annually by up to ~2 Sv. There is no observed trend in Sverdrup or Ekman transports. As considered in previous studies (e.g., Marsh et al. 2017), eastward Ekman transport at the shelf break makes only a minor contribution to Slope Current transport, being most substantial during strong

southwesterlies in winter. The sea surface slope indicative of surface geostrophic flow at 30 °W (Fig. 6c) varies in synchrony with the geostrophic and total transport (Fig. 5), dominated by decadal variability superimposed on long-term decline.

Figure 7 shows the same GODAS absolute transports respectively, binned in temperature and salinity space (in intervals of 0.5°C and 0.05 respectively) and shown on a Temperature-Salinity (T-S) diagram for (a) pre-1997 and (b) 1997 onwards, to further understand the observed shift to warmer transport from a water mass perspective. Transport is spread across temperatures in the range 5 - 22 °C and salinities in the range 34.25 – 36 pre-1997 (Figure 7a). Post-1997 (Figure 7b) shows a similar temperature distribution but is more constrained in salinity, with a range of 34.7 – 36.2. The corresponding potential density range is 26.8 – 27.0 kg m$^{-3}$ (GODAS), with higher values corresponding to Subpolar Mode Water (McCartney and Talley, 1982). A small amount of negative (westward) transport is observed at temperatures below 7.5 °C in both analyses. The strongest transports occur at temperatures of 7.5 – 10 °C and 35.25 – 35.50 in GODAS: this T-S class transport strengthens by at least 0.02 Sv, but overall transport determined by the sum of the gridcells has decreased. Differences in pre-1997 and post-1997 transport distributions in T-S space are consistent with the declining total transport and increase in the significance of the warmer flow observed in Figure 5. An overall regime shift of higher salinities and temperatures is observed in the post-1997 period, with a decrease of total transport. Once again, this is consistent with the time series shown previously. There is a greater spread of temperature from 1997, however a narrower salinity range is apparent.

### 3.3 Shelf edge transport and Lagrangian evidence for changing provenance of Slope Current water

Shelf edge current velocity from the ORCA12 model hindcast was plotted in Figure 1, with quivers showing the magnitude and direction, and the background colour indicating the northward velocity component. 245m was the chosen model depth level to target the high-velocity Slope Current core and negate the effects of large flows at the sea surface caused by surface heating and winds. This depth level also resides within the accepted range of the Slope Current core (Porter et al., 2018). Northward velocity has clearly decreased between 1992 (Figure 1 panels a, winter; and c, summer) and 2010 (Figure 1 panels b, winter; and d, summer). Winter flows appear stronger in both years. To examine how eastward transport in the SPG region of the North Atlantic translates to northward volume transport at the shelf edge (unresolved in GODAS), four approximately cross-slope (zonal) transects were identified in the ORCA12 hindcast, at 53, 56, 57 and 58 °N (Figure 8, with the transects annotated on Figure 1). At each transect, transport was calculated from 5-day averages of the northward component of velocity, in 0.5 °C temperature bins. As for the zonal transports at 30 °W presented earlier, we partition shelf-edge meridional transports above and below 11 °C. These transports and corresponding annual means are presented in Figure 8. Transports across 56-58 °N (Fig. 8a-c) are up to 5 times greater than the transports at 53 °N (Fig. 8d), consistent with recruitment of zonal inflow to the Slope Current along the Hebridean shelf (Porter et al., 2018). Given considerable shelf edge exchange poleward of 56 °N, and the choice of transects that may more or less capture the Slope Current flow, transports across 56-58 °N are consistent with progressive inflow from the west: the shelf edge transport time series show similar variability, peaks and troughs to the GODAS-derived geostrophic and total eastward transports in Figure 5. Notable in these time series is considerable variability, on timescale from synoptic to decadal. In particular, transport at 56 - 58 °N declines from the mid-1990s onwards, contemporaneous with the declines in zonal transport evident in Figure 5. Prior to 1997, annual-mean transports at 56-58 °N remain above 3 Sv. Post-2001, annual-mean transport at 58 °N does not exceed 2 Sv and falls below 1.5 Sv at 56-58 °N during most years over 2007-10. Transports across 53 °N reveal a more distinct seasonal cycle, with peak transport each winter, but little of the long-term decline seen further to the north.

The temperature partitioning emphasises a sharp contrast between 53 and 56 °N. Almost all transport at 53 °N is in temperature classes exceeding 11 °C at all times of the year, interrupted by occasional cold pulses. Transports across 56-58 °N is dominated by temperature classes below 11 °C, with transports of water warmer than 11 °C restricted to late summer and early autumn in most years, when this water can briefly account for the majority of transport. On the longer timescale, this fractional contribution of warm water to Slope Current transport across 56-58 °N is relatively steady, in spite of the decline in total transport. We can therefore deduce that the weaker transport in later years is warmer. This is the same as we observed in the eastward transports from the subpolar North Atlantic in Figure 5.

To trace inflows to the Slope Current, Lagrangian 4-year hindcast experiments were run: releasing virtual water "particles" in 1992 and 2010 at the core of the Slope Current (see Sect. 2.4 for details). These dates were chosen to obtain the flow pathway of water towards the shelf edge before and after the 1997 regime shift as observed in the analysis of the GODAS data (as shown in Sect. 3.2) and noting the model trends in shelf-edge transport that were previously highlighted. Figure 11 shows the differences between the two ARIANE particle tracking 4-year hindcast experiments (Figure 9 and Figure 10): the ensemble mean output for 1992 was subtracted from the 2010 ensemble means. The output, specifically the fractional presence of particles (panel a in Figs. 9 - 11) shows a shift – comparing later to earlier backtracking – to a more south-westerly origin of particles ending up at the shelf edge. The vast majority of particles originate off the eastern coast of the United States, with the most followed pathway corresponding with the Gulf Stream and North Atlantic Current. This is also the fastest pathway, with particles off the US east seaboard having a mean age of ~600 days (Table 1). A smaller fraction of particles can be traced back to the subpolar gyre. The later experiment also shows more influence of water traced to the Bay of Biscay. The mean particle depth subplots (panel c in figs. 9 - 11) indicate that most particles originate from depths of approximately 100-150m. Only outside of the NAC and SPG pathway are particles traced to greater depths. The difference plot shows that overall, the 2010 hindcast saw particles become deeper by up to 100m (Figure 11c), particularly the particles of subtropical origin. Flows were slower by approximately 100 days, mainly when following the NAC pathway. Comparing 2010 to 1992, there has also been a considerable increase in upstream temperature (ranging from 1 – 2 °C) of particles arriving at the shelf edge. This is confirmed in the T-S analysis (Figure 7) and in the summary statistics presented in Table 1.

We can track particles passing a selected meridian: 30 °W is again been chosen, due to the high prevalence of particles crossing that longitude in both hindcasts. Figure 12 and Figure 13 show histograms of the two hindcasts (1992 and 2010), where particles are tracked passing 30 °W in the latitude range 40-60 °N. Figure 12 shows the latitude (panel a for 1992, c for 2010) and time (particle age, panel b for 1992, d for 2010) at which they crossed. Figure 13 shows the temperature (panel a for 1992, c for 2010) at which they crossed, as well as the depth of crossing (panel b for 1992, d for 2010). Only the first crossing (since release of the hindcast) of 30 °W is recorded in both figures. The latitude at which the particle first passes 30 °W has shifted south overall, as shown in Figure 12e. The statistics shown in Table 1 confirm this visual analysis, with the median shifting southward by 1.2 degrees. These observations are consistent with the long-term transport changes, both geostrophic and absolute (Figure 5). Even more striking is the change of particle age (Figure 12f). The 2010 hindcast shows a median slowdown of 115 days. This confirms the general pattern evident in the ensemble mean difference plot (Figure 11). The latitude of its first crossing of 30 °W shows a normal distribution in both hindcasts. However, the spread of the latitude at which the particles crossed increased by 0.32°, measured by standard deviation (table 1). The spread of particle temperatures (Figure 12e) and particle age (Figure 12f) has also increased in the 2010 hindcast.

## 4 Discussion

As determined in a wide range of previous studies, our results have confirmed the North Atlantic is warming and transport is changing over a decadal timescale, despite being highly variable interannually and spatially. We first discuss the evidence for physical and dynamical changes in the North Atlantic and associated changes in the Slope Current, followed by consideration of the potential implications of these changes.

### 4.1 Variable hydrography and transport

Four decades of GODAS data have shown that the hydrography of the North Atlantic has changed considerably. Variation in density anomaly appears to be influenced more by temperature change than of salinity, due to the close inverse relationship between temperature and density anomalies (not shown). Until recently, the subpolar gyre was subject to long-term warming that began around 1997. In the mid-2010s, the re-establishment of positive density anomalies was associated with dramatic cooling in the SPG region, with the core centred at 40 °W, 47 °N. This is largely associated with extensive cooling of the SPG from 2013 – 2015, which saw anomalies of up to -2.5 °C extending to at least 600m (Josey et al., 2018). This was then termed the "Big Blue Blob" (or "Cold Blob") by some within the field and indeed the media (Mooney, 2015;Josey et al., 2018).

Associated with basin-scale changes in hydrography are anomalies at the eastern boundary, expressed in density along the continental slope. Plotting density anomalies in time and latitude (Figure 4), there is meridional coherence of anomalies consistent with conveyance along the Slope Current pathway, although pulses of positive or negative anomaly appear to originate from the south and weaken poleward in some years. The early 2000s show a shift from an overall negative temperature anomaly to a positive anomaly (not shown), consistent with the decadal mean anomaly maps in Figure 2. Emphasizing the subtropical gyre as a driver of interannual variability, although restricted to analysis over 1992-2002, (Pingree, 2002) finds that poleward flow at the eastern boundary is enhanced following winters when the NAO index was negative. In contrast, we find here that Slope Current variability on a longer timescale is attributed to changes in the subpolar gyre that drive variations in mid-latitude flows towards the eastern boundary.

Examining large-scale inflow towards the eastern boundary, geostrophic and total eastward transport estimates obtained with GODAS (Figure 5), show an overall decline in transport of approximately 5 Sv throughout the time series. This variability is clearly associated with changes of density and cannot be directly attributed to wind forcing (from Sverdrup balance). Partitioning water warmer or cooler than 11°C, we identify an increase in warm-water (southern) inputs to the Slope Current since late 1990's. This has been observed in the transport calculations as well as the ARIANE particle tracking (Figure 9 – Figure 11). The particle tracking calculations provide evidence of a southward shift of water feeding into the Slope Current over the past 3 decades by nearly 1° of latitude (Figure 11). This is consistent with the observed southward transport shifts seen in the SPG region (Figure 5 and Figure 6). Pulses of water near the shelf edge have been observed before: transport anomalies of +4 Sv were observed through the Rockall Trough, corresponding to periods of increasing Atlantic input to the North Sea (Holliday and Reid, 2001). Plotting GODAS transports in TS space (Figure 7) confirms the shift towards warmer (and more saline) transports. A warming and more saline trend has previously been observed for shelf-edge flows (Berx et al., 2013), although at the time the authors did not determine any trends in volume transport.

Our results have shown that density gradient driven transport (Figure 5a) correlates strongly with northward transport at the shelf edge, and we conclude that this is the main driver of variability in Slope Current flows. Whilst wind stress and associated Ekman transport is significant in forcing local short term shelf edge and North Sea transport (Pätsch et al., 2020),

there is no evidence for a long-term trend in either Sverdrup or Ekman transport. That SSH gradient anomalies closely resemble those in eastward transport time series is consistent with the steric effect of density anomalies. Principal component analysis of SSH has accordingly been used as an index of the subpolar gyre strength (Hátún and Chafik, 2018).

**4.2 Implications of the observed changes on the Slope Current and European Shelf Seas**

Changes in the meridional density gradient across mid-latitude North Atlantic drives changes in geostrophic eastward transport that subsequently feeds the shelf edge northward flow (Marsh et al., 2017). Corresponding changes in the strength and properties of the Slope Current are likely to reverberate downstream. Expansion and contraction of the subpolar gyre has already been shown to be correlated with salinity variations on an interannual and decadal timescale in the northern North Sea. strongly linked to pulses in nutrient availability, with a lag of approximately 2 years (Hátún et al., 2021;Hátún et al., 2017;Jacobsen et al., 2019). Nitrate and phosphate fluxes in the North Sea have been linked to the local and regional wind fluxes (Pätsch et al., 2020). The SPG has been proposed as a regulator of silicate concentrations on the central Faroe shelf (Pätsch et al., 2020). In support of this, we emphasize the strong link established here between eastward transport indices in the subpolar North Atlantic (Figure 5 and Figure 6) and northward transport at the shelf edge (Figure 8).

The Slope Current acts to convey water with distinct Atlantic temperature and salinity, as well as nutrients from the North East Atlantic along the shelf edge and eventually into the North Sea (Reid et al., 2003;Porter et al., 2018). Water from the Slope Current is also upwelled to the surface and transferred to the shelf seas in the Whittard Canyon, south west of Ireland (Porter et al., 2016). The amount of cross-shelf exchange varies seasonally and interannually and can switch to a downwelling mode depending on the strength and direction of Slope Current flow (Porter et al., 2016). This provides an important mechanism for drawing nutrients such as nitrate, phosphate, and silicate from deeper flows onto the shelf (Porter et al., 2016), making them available for biological consumption. Changes in the Slope Current may consequently impact downstream ecosystems through a number of mechanisms. There is growing evidence that recent warming trends have impacted shelf sea species distributions, leading to "subtropicalization" of the North Sea: warmer-water species of zooplankton and fish species have been observed (and in some cases, now breeding) in UK coastal waters (Montero-Serra et al., 2015;Beaugrand et al., 2009;Beare et al., 2004). Not only is the species distribution being altered, the warmer water is physically affecting fish such as cod with "heat-induced hyperglycaemia" and water oxygen saturation is decreased with the warming water (Beaugrand et al., 2008). With up to 40% of Slope Current flow destined for the North Sea (Marsh et al., 2017), changes to the provenance and properties of the current could have profound effects to the state of the northern North Sea in particular, and may provide the conduit for the observed "subtropicalization". Whilst the species shift has not been explored in this study, it clearly warrants further work to assess the significance of Slope Current variability on North Sea inputs and ecology.

**5 Conclusions**

We have shown that broad warming of the subpolar North Atlantic Ocean, with a mean of around 1 °C between our two study hindcasts (1992 and 2010), has acted to decrease the meridional density gradient of the region, leading to a general reduction and slowdown in geostrophic transport (by up to 10 Sv) feeding the Slope Current. More recent extreme cooling events of up to 2 °C have also been a major feature, but these have been relatively short lived. Transport towards the shelf edge has been decreasing since 1995-97, although some evidence exists of a slight recovery post-2015. It remains too early to tell if this recovery will persist. Shelf edge northward transport, incorporating the Slope Current, shows a similar pattern

but with a smaller magnitude of 2 Sv north of the Rockall Trough. This relationship supports the theory that a major input to the Slope Current is subpolar gyre water. There has been a gradual, sustained southward shift (of approximately 1° of latitude) in the water flowing to the shelf edge and being incorporated in the Slope Current. Back-tracked from 2010, transport times towards the shelf edge from 30 °W have increased by an average of 118 days, based on comparison between our two particle back-trajectory ensembles. The spread (standard deviation) of particle latitudes, temperatures, ages and depths at time of crossing 30°W have all increased, which is associated with the weakening eastward flow from the subpolar North Atlantic.

There are yet unquantified implications of these changes, with particular focus on subtropicalization and nutrient inputs to the North Sea and the rest of the UK/European shelf. Quantifying the effects of the changing Atlantic inputs to the Slope Current and surrounding shelf seas will be examined in further work.

**Acknowledgements and funding**

We would like to thank the Natural Environment Research Council for funding this work through the SPITFIRE DTP (grant number: NE/L002531/1). We also thank the National Oceanography Centre for the use of their computer systems and allowing access to the ORCA12 model outputs. Thanks also to Science and Technology Facilities Council (STFC) and the Centre for Environmental Data Analysis (CEDA) for allowing access to the JASMIN computer systems. Finally, we wish to thank our reviewers Hjálmar Hátún and an anonymous second reviewer for their constructive comments on our manuscript.

**Competing interests**

The authors declare that they have no conflict of interest.

**Code and Data Availability**

GODAS and NCEP data provided by the NOAA/OAR/ESRL PSD, Boulder, Colorado, USA, freely available from their website at https://www.esrl.noaa.gov/psd/. ARIANE Lagrangian particle tracking software was freely provided by B. Blanke and N. Grima, Laboratoire de Physique des Océans, France. ORCA12 data is provided on request to the National Oceanography Centre, UK. Code used for data analysis and plotting in this project is available via the public repository service "Zenodo" (Clark et al., 2021), doi: 10.5281/zenodo.4965880 Data analysis and plotting of Figs. 1 – 10 was performed using Python 2.7 in a Spyder environment (environment file is provided in the Zenodo repository). Figs. 11 – 15 use Matlab script coded in Matlab 2017a.

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

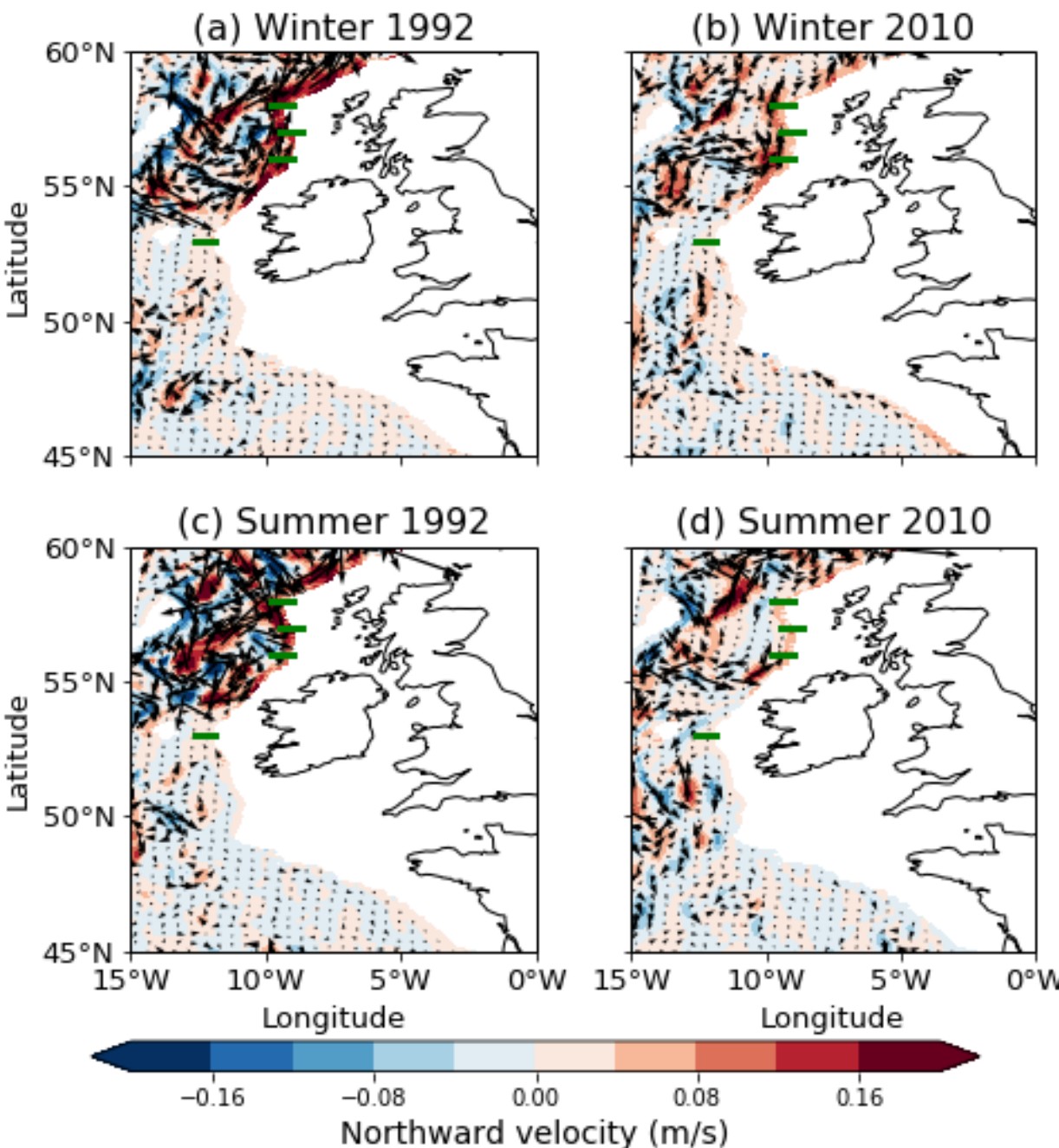

**Figure 1: quiver plot, using ORCA12 data, indicating subsurface (245 m) velocity magnitude and direction of the water, comparing winter (January) and summer (July) releases: (a) winter 1992; (b) winter 2010; (c) summer 1992; (d) summer 2010. Background colour highlights the magnitude of the v component of velocity, positive northwards. Green lines indicate the zonal transects used for Figure 8.**


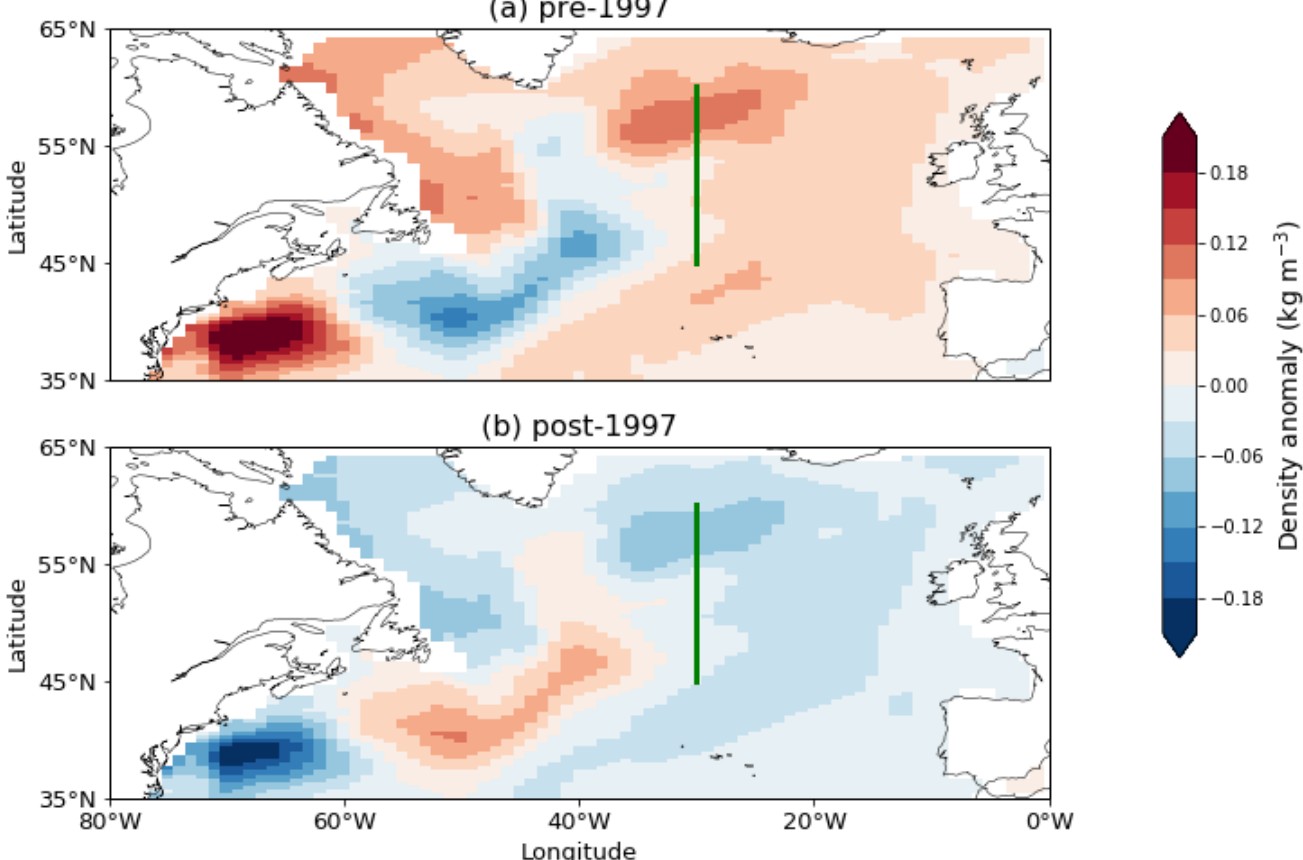

**Figure 2: Mean anomaly maps of density at 205 m below the surface: (a) pre-1997; (b) post-1997. Data from GODAS. Anomalies calculated with respect to the 1980 – 2020 climatology. Green line at 30°W shows the meridional profile used for eastward transport calculations, between 45 °N and 60 °N**

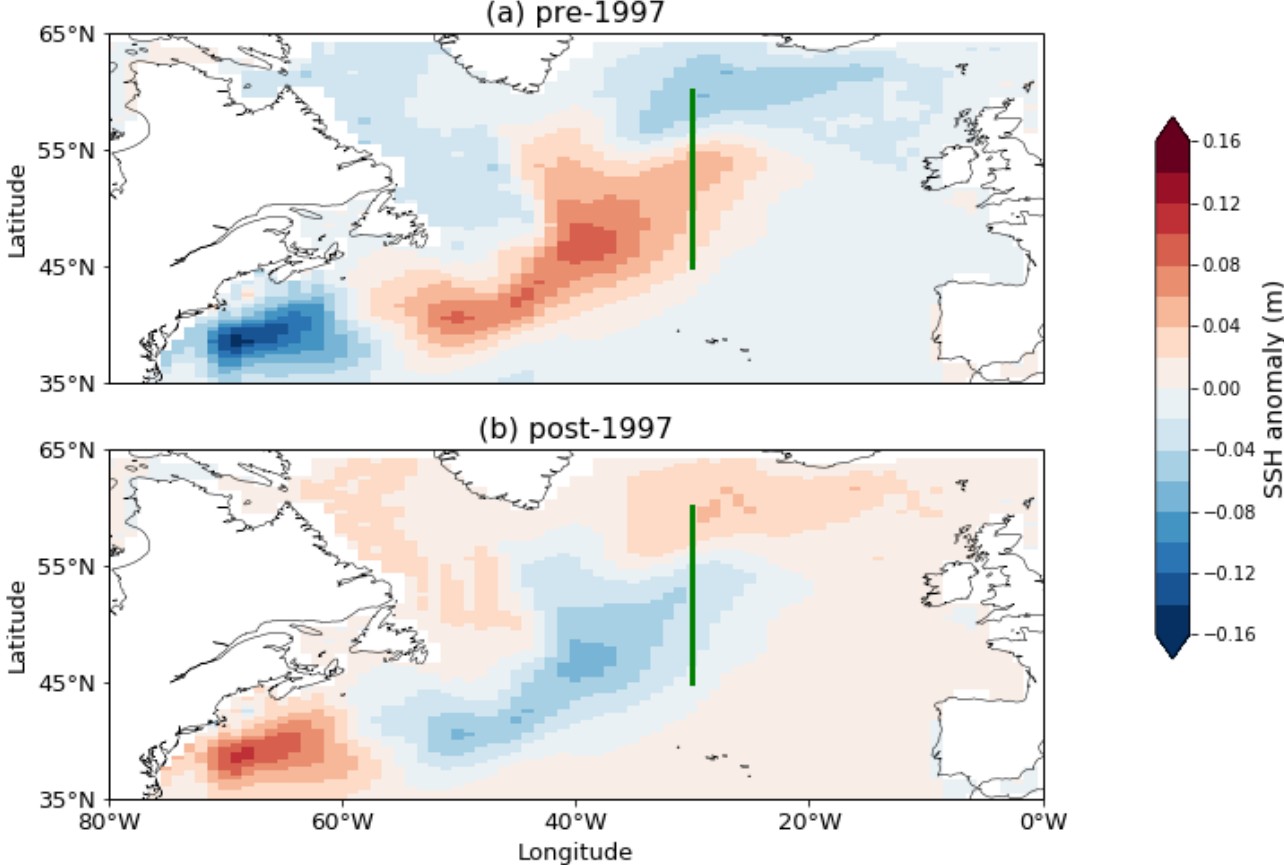

**Figure 3: Mean anomaly maps of sea surface height (SSH): (a) pre-1997; (b) post-1997. Data from GODAS. Anomalies calculated with respect to the 1980 – 2020 climatology. Green line at 30 °W shows the meridional profile used for eastward transport calculations, between 45 °N and 60 °N.**

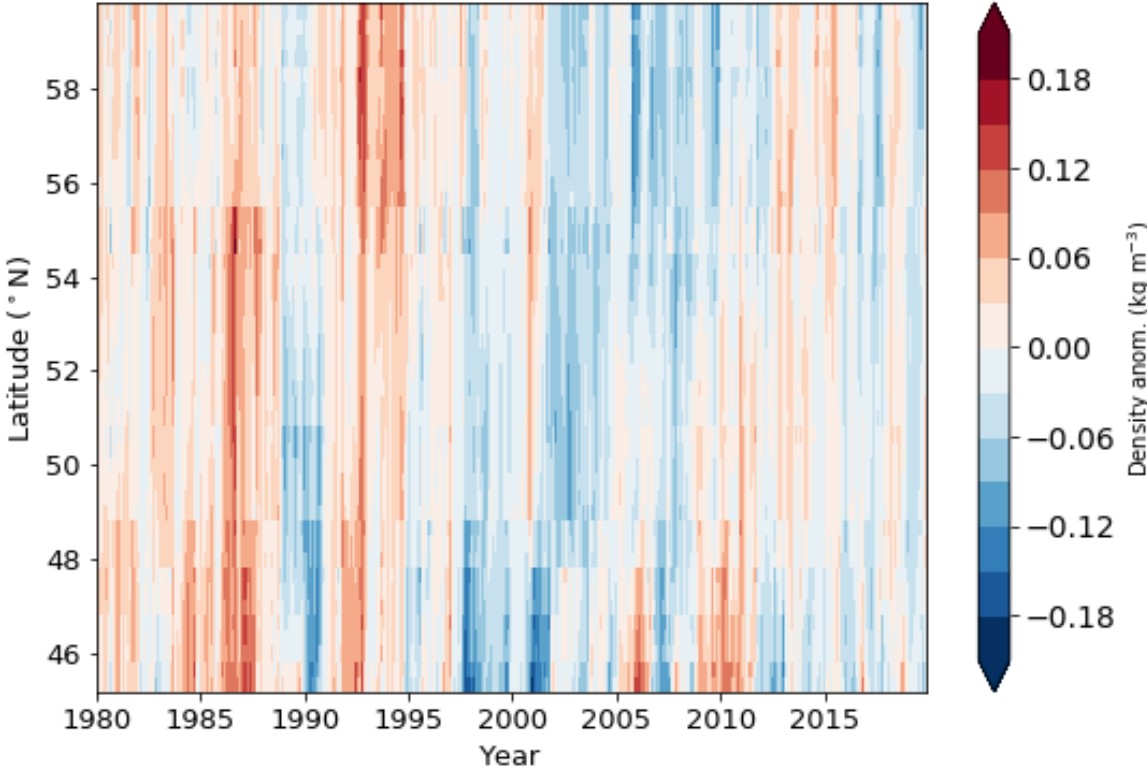


**Figure 4: Time series Hovmöller plot of GODAS mean density anomaly, averaged over depth levels 0 – 600 m, at the shelf edge between 45 – 60 °N.**

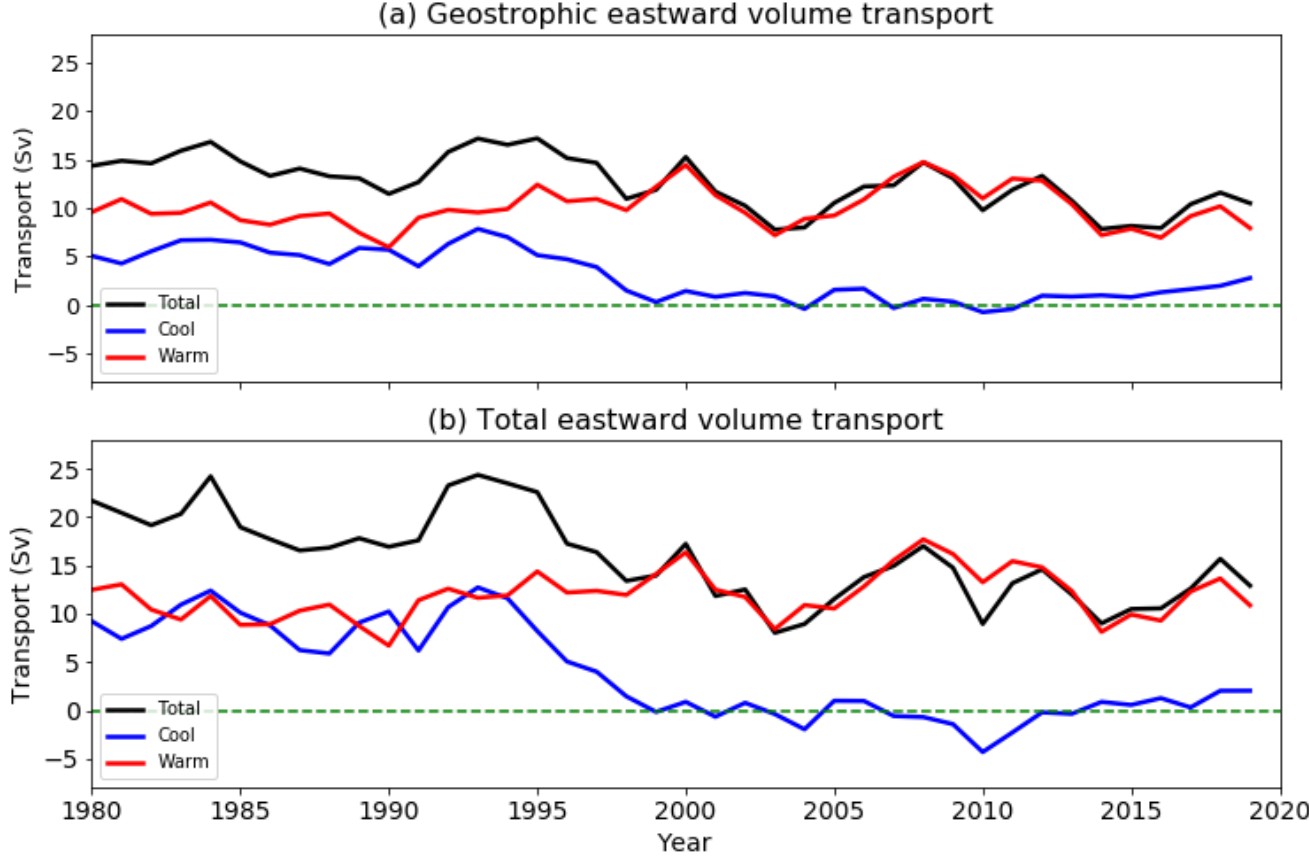


**Figure 5: Eastward volume transport time series (annual means) from the GODAS dataset, (a) Geostrophic; (b) Total; at 30 °W between 45.2 °N - 60.2 °N and in the upper 1000 m. "Cool" is defined here as temperatures < 11 °C, "warm" is ≥ 11 °C.**

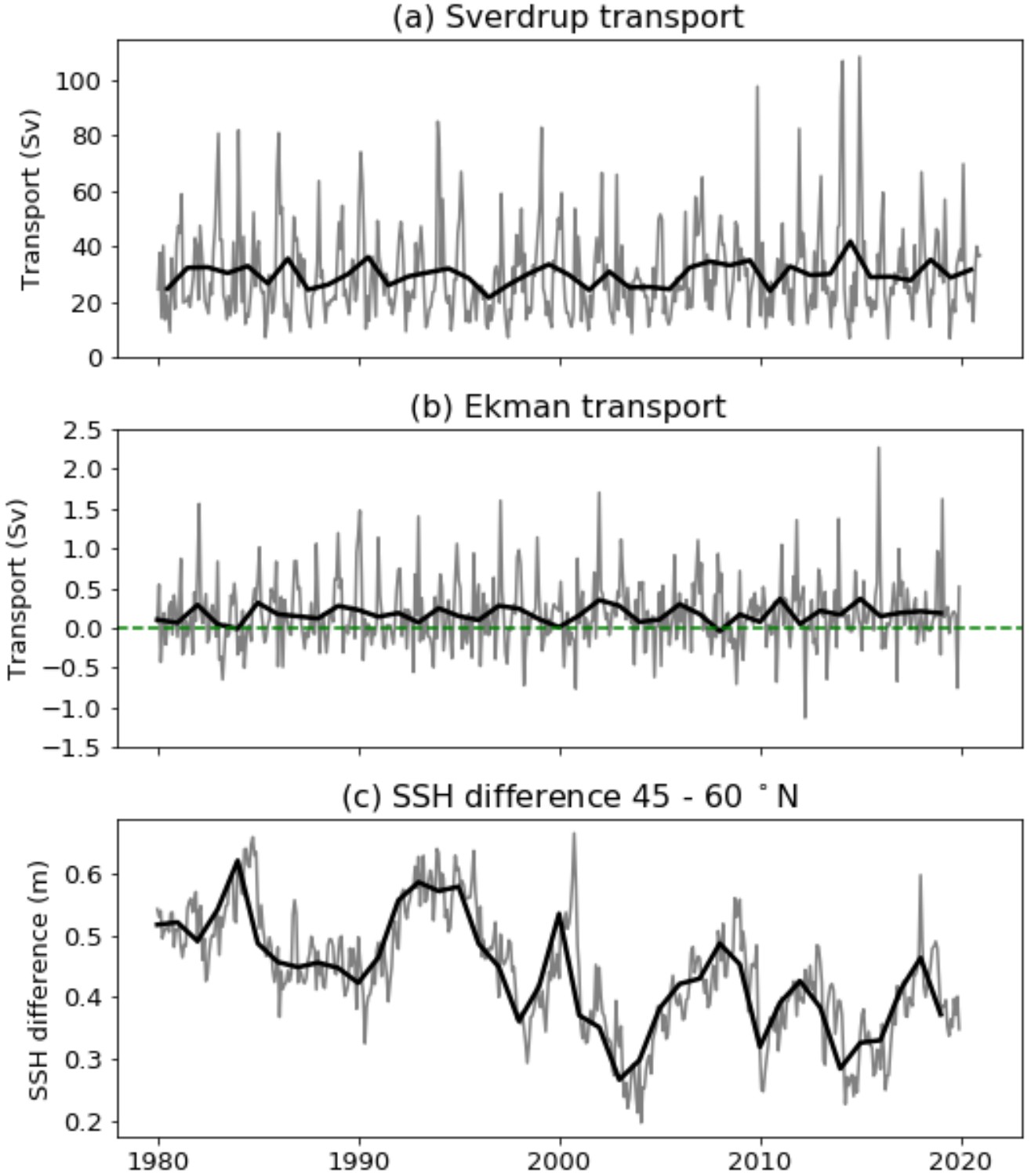

**Figure 6: Annual and monthly mean eastward volume transport time series: (a) Sverdrup transport, calculated from NCEP winds; (b) Ekman transport, calculated from GODAS northward momentum (wind) flux, for the shelf edge region (15 °W, 50 – 58 °N); (c) SSH difference between 45 °N and 60 °N, at 30 °W**

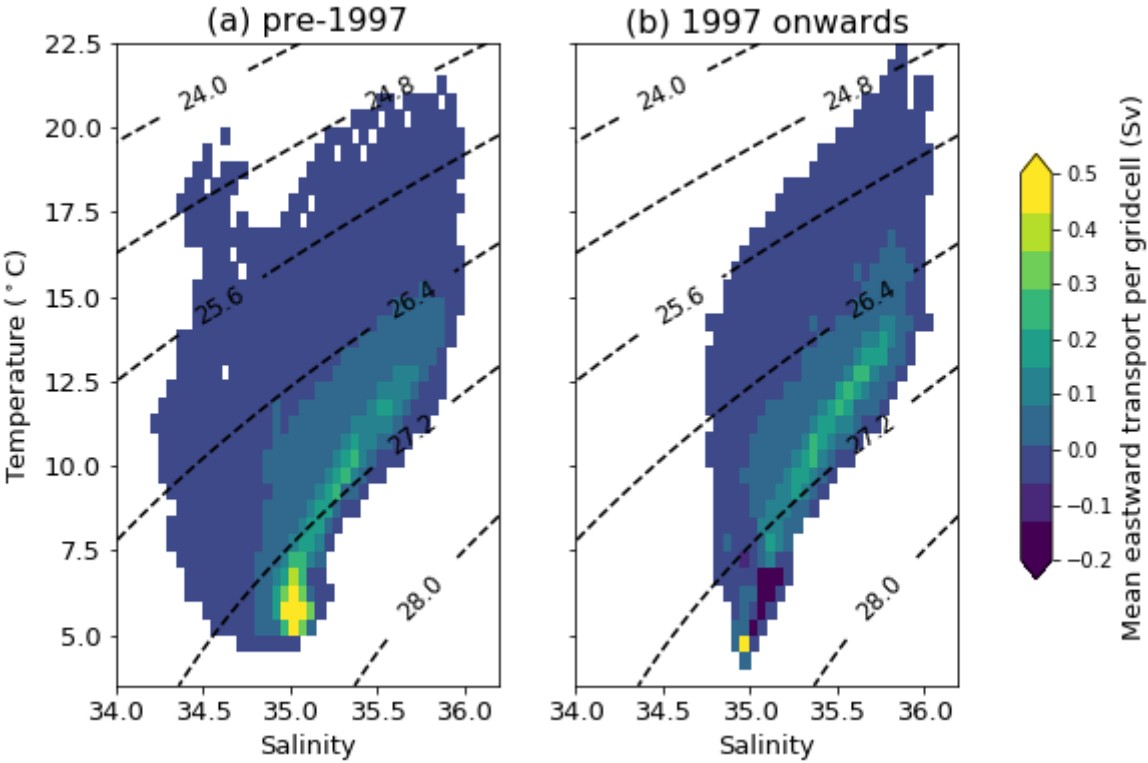

Figure 7: GODAS Temperature-Salinity binned eastward transport at 30°W between 40 – 58 °N, in the upper 1000 m for (a) pre 1997 and (b) 1997 onwards. Dashed lines are isopycnals. The sum of the gridcells in (a) is 17.2 Sv and (b) 11.1 Sv.

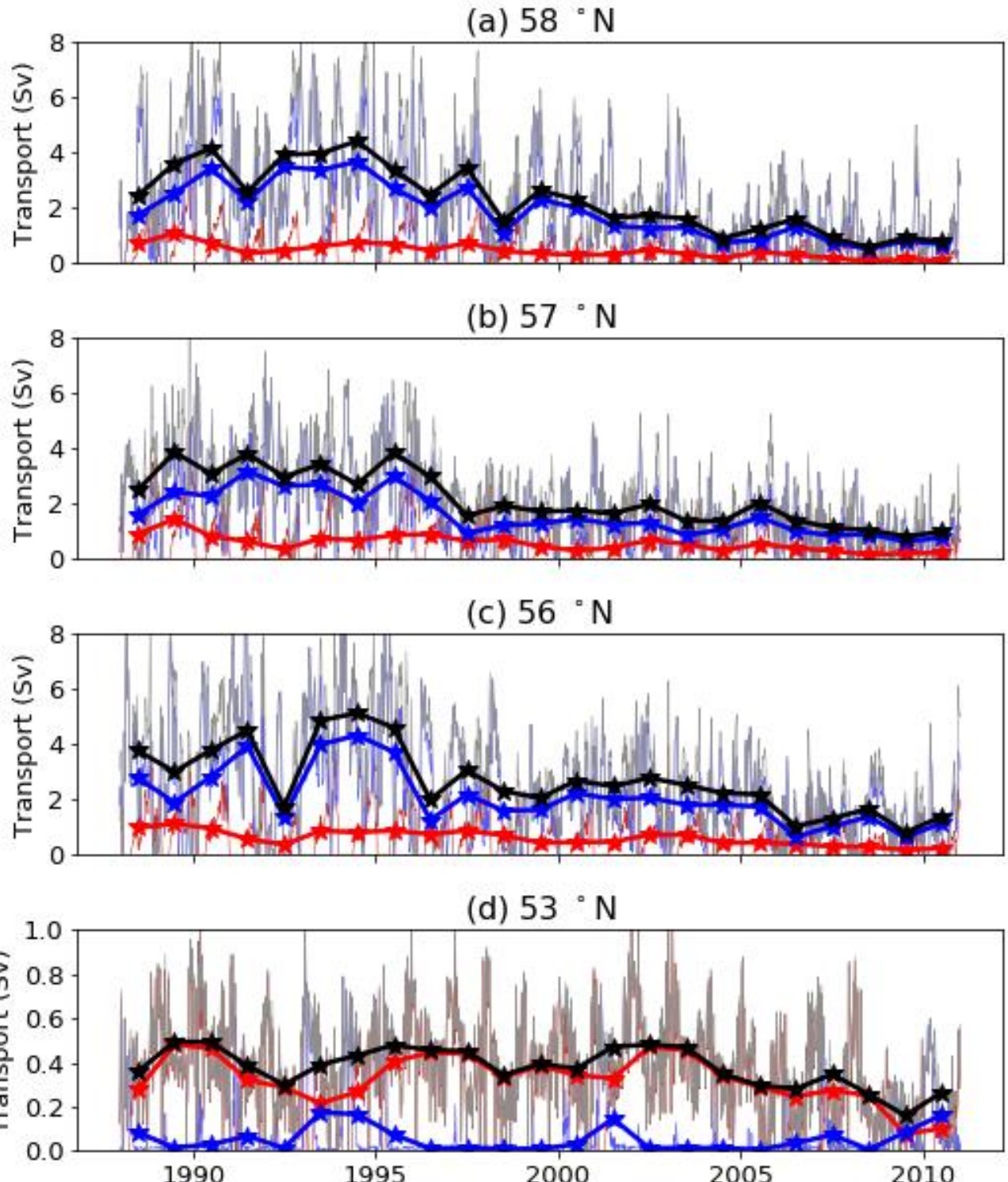

**Figure 8: Northward total transport at the shelf edge from the ORCA12 hindcast, across zonal transects at (a) 58°N, (b) 57°N, (c) 56°N, and (b) 53°N (see Fig. 1). Transects are the following length, as constrained by the bathymetry: 53°N is 53.7 km; 56°N is 51.0 km; 57°N is 50.2 km; 58°N is 49.5 km. Total transport is indicated with black lines. Transport at temperatures above and below 11°C is indicated with red and blue lines respectively. Thin lines show 5-daily means. Thick lines and stars indicate the annual mean. Note that Y-axis scales on (d) are different from (a)-(c).**

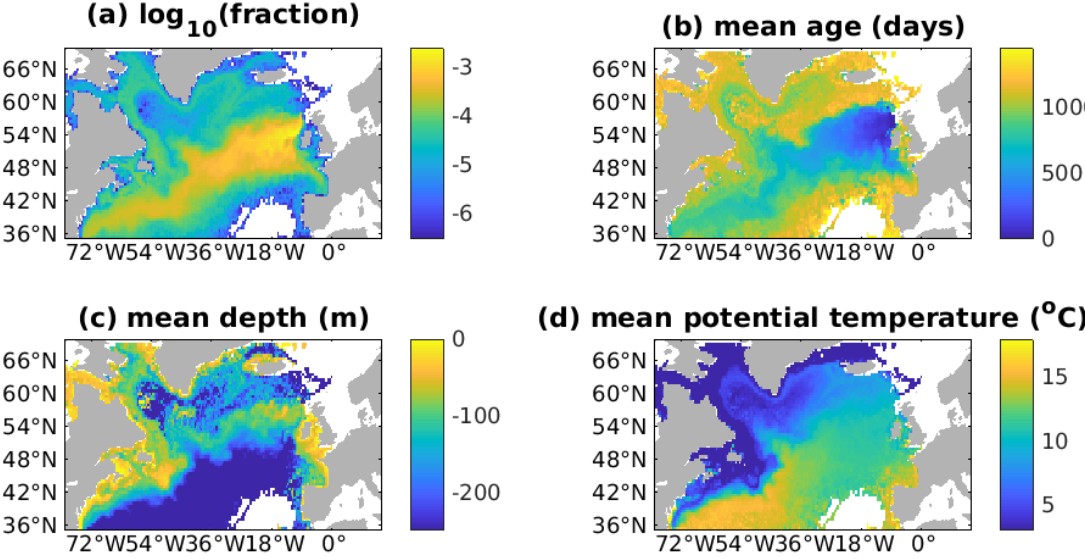

**Figure 9: Ensemble mean particle statistics for the 1992-88 Ariane hindcast: (a) log$_{10}$ fraction of total particles; (b) mean age in days; (c) mean depth in metres; (d) mean potential temperature in °C**


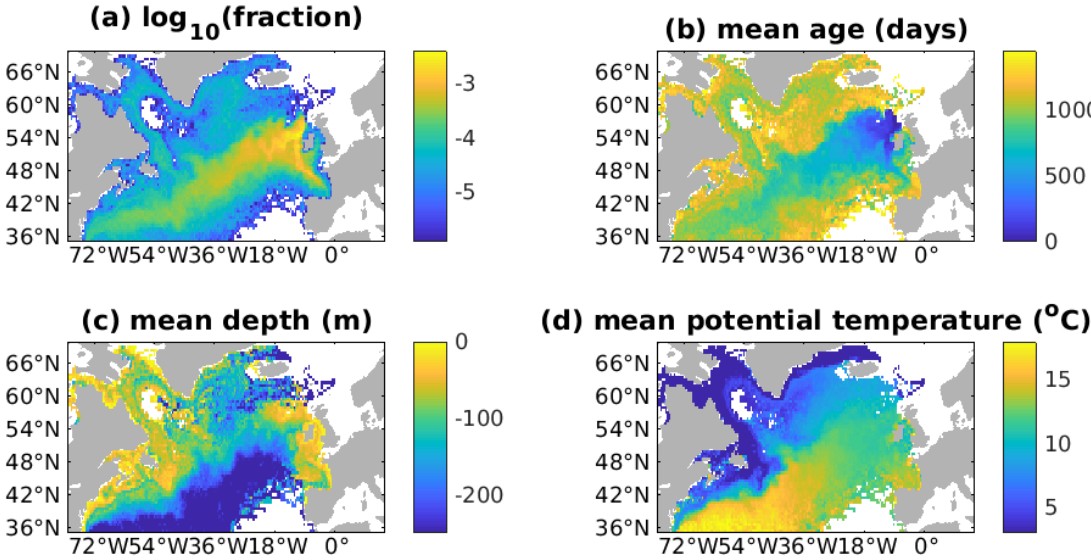

**Figure 10: Ensemble mean particle statistics for the 2010-06 Ariane hindcast: (a)** $\log_{10}$ **fraction of total particles; (b) mean age in days; (c) mean depth in metres; (d) mean potential temperature in** °C

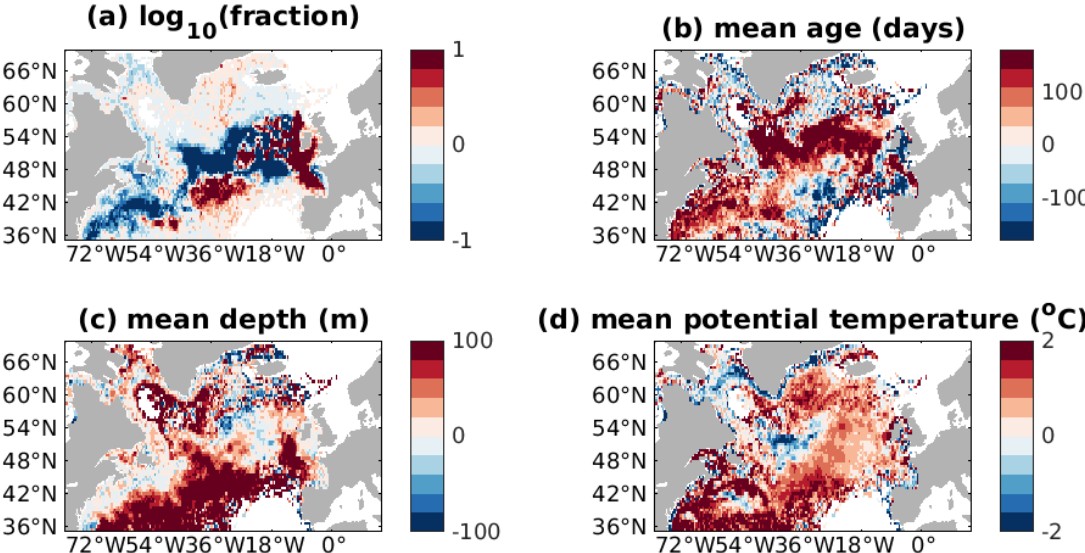

**Figure 11: Difference between the ensemble mean statistical analyses of the ARIANE 4-year hindcasts: (a) 2010 mean – 1992 mean log₁₀ fraction of total particles; (b) 2010 mean – 1992 mean age in days; (c) 2010 mean – 1992 mean depth in metres, positive = shallower; (d) 2010 mean – 1992 mean potential temperature in °C.**


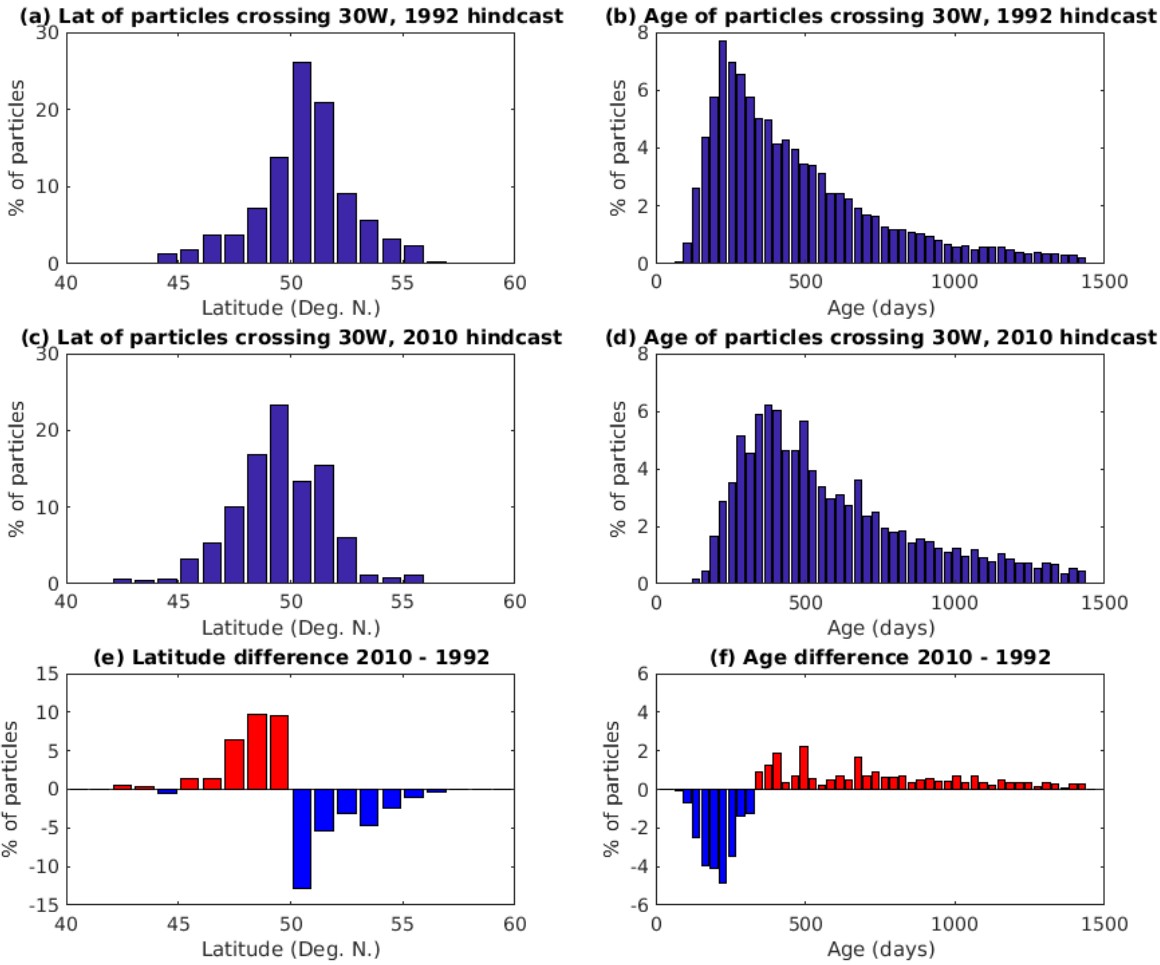

**Figure 12: Histograms from the 1992 and 2010 hindcast, showing the latitude and age of unique particles crossing 30 °W for the first time (any further crossings are ignored). (a) 1992 particle latitude; (b) 1992 particle age; (c) 2010 particle latitude; (d) 2010 particle age; (e) latitude difference; (f) age difference. Positive difference = more particles in 2010. Data binned by 1° latitude and 30 days.**


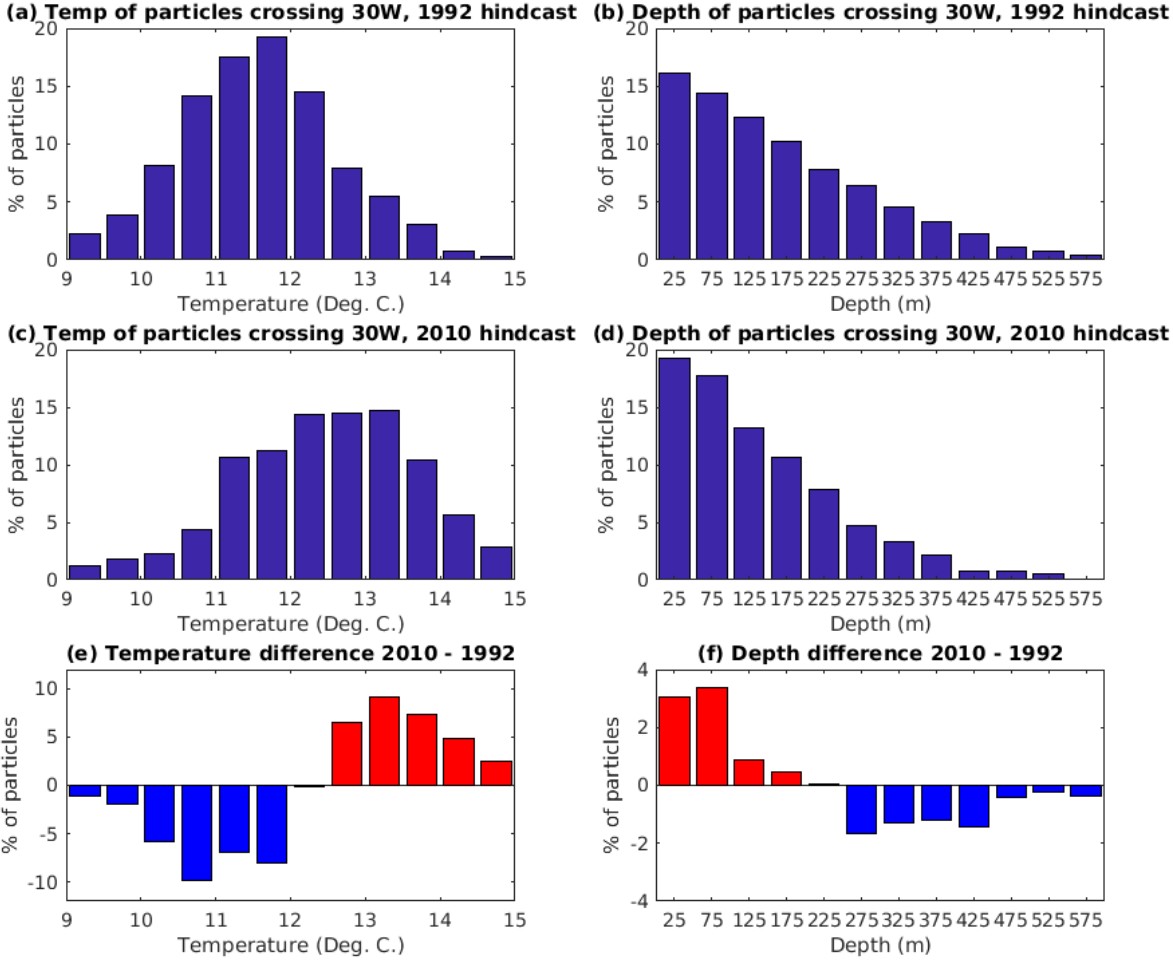

**Figure 13: Histograms from the 1992 and 2010 hindcast, showing the temperature and depth of unique particles crossing 30 °W for the first time (any further crossings are ignored). (a) 1992 particle temperature; (b) 1992 particle depth; (c) 2010 particle temperature; (d) 2010 particle depth; (e) temperature difference; (f) depth difference. Positive difference = more particles in 2010. Data binned by 0.5 °C temperature and 50m depth.**

**Table 1: simple statistics to assess the spread of and difference between the two Ariane hindcasts**

| Statistic | 1992 hindcast | 2010 hindcast | Difference 2010 – 1992 |
|---|---|---|---|
| Latitude: Mean of crossing 30 °W | 51.68 °N | 50.73 °N | -0.95 °N |
| Latitude: Median of crossing 30 °W | 51.72 °N | 50.44 °N | -1.28 °N |
| Latitude: Standard deviation of crossing 30 °W | 2.34 ° | 2.66 ° | 0.32 ° |
| Age: Mean of crossing 30 °W | 498 days | 616 days | 118 days |
| Age: Median of crossing 30 °W | 420 days | 535 days | 115 days |
| Age: Standard deviation of crossing 30 °W | 278 days | 289 days | 11 days |
| Depth: Mean of crossing 30 °W | 177.8 m | 156.9 m | -21 m |
| Depth: Median of crossing 30 °W | 147.6 m | 133.1 m | -14.5 m |
| Depth: Standard deviation of crossing 30 °W | 138.7 m | 116.9 m | -21.8 m |
| Temperature: Mean of crossing 30 °W | 12.0 °C | 12.9 °C | 0.9 °C |
| Temperature: Median of crossing 30 °W | 12.0 °C | 13.1 °C | 1.1 °C |
| Temperature: Standard deviation of crossing 30 °W | 1.2 °C | 1.6 °C | 0.4 °C |