# Peer review of "Weakening and warming of the European Slope Current since the late 1990s attributed to basin-scale density changes."

_Ocean Science, 2021_

## Author Response (AR1)

*December 2021*

Once again, we would like to thank both reviewers for taking the time to review our manuscript. We have taken on board the feedback and have made alterations to try and satisfy their comments and criticisms. Here, we present point-by-point responses to the reviewer comments, outlining what amendments we have made. As before, we will use italics to show the original reviewer comments.

**Firstly, we consider Reviewer 1's comments (Hátún):**

*The discussion on eastward flows in the North Atlantic Current has a long history (which the present study does not appear to acknowledge). Already in 2002, Pingree (2002) linked this to the meridional gradient in sea surface height (SSH), as revealed by satellite altimetry. He discussed increased SSH gradient/transport during NAO+ years (e.g. 1994-1995 and 1999-2000). He (and a large volume of following literature) has associated such interannual changes to the NAO index and variability in the wind stress curl field over the NE Atlantic.*

*After this, SSH data (observed and simulated) have been utilized by myself and many others, to discuss this dynamics in a broad and longer context (Hátún and Chafik, 2018, and reference therein). For example is the calculation of the so-called gyre index closely linked to variability mentioned by Pingree (2002), and likely to your eastward transport records based on hydrography. Your analysis on these transports is interesting, and does provide new information/knowledge. Just try to better weave it into the existing volume of knowledge. This includes paying more attention to the interannual fluctuations that you present (Figure 6 and 7). Your study is clearly motivated by identifying drivers behind ecosystem fluctuations along the European continental slope (ECS). We have previously linked these pulses to many ecological aspects in the NE Atlantic (e.g. Hátún et al., 2017, 2016; Jacobsen et al., 2019), and a growing body of evidence shows that this type of variability does also characterize the ECS (Pätsch et al., 2020). You have the evidence, utilize it better.*

We have now added further background information about SSH dynamics in the North Atlantic into the introduction (lines 59 – 64), and now relate our findings to previous satellite altimetry studies (including the papers mentioned by Hátún) in section 4.1. We have also analysed GODAS SSH: plotting mean anomaly maps of SSH pre and post 1997 (Fig. 3), and using SSH meridional gradients at our study transect (30 °W, 45 - 60° N) as a proxy for the barotropic eastward transport (Fig 6c). In relation to the SSH proxy, we further note 'That SSH gradient anomalies closely resemble those in eastward transport time series is consistent with the steric effect of density anomalies.' (lines 401-402).

On Hátún's comments on our motivations to study the drivers of ecosystem fluctuations, we have better related our findings to the literature suggested (amongst others) in section 4.2

*Garcia-Soto et al. (2002) and Pingree (2002) discuss the conditions along the ECS in relation to the relatively narrow slope currents from the south (Bay of Biscay). This topic should be better handled in your work. For example does Pingree (2002) claim that the North Atlantic Current strength and the mentioned poleward flow are out of phase. Weak NAC (aka NAO-) is related to stronger flow of warm and saline waters from Spanish waters – also referred to as Navidad events/years (Garcia-soto, 2002). This seems to be at odd with your perspective (although I follow your argument that NAC waters are being continuously recruited to the slope, north of the Porcupine Bank). This aspect must be better handled.*

We now present the findings of Garcia-Soto et al (2002) in the introduction (lines 64-66) and attempt to relate our findings to the NAO in the discussion.

*Fig. 1 does nice illustrate the entrainment of water to the boundary north of Ireland, and no northward bound boundary current south of the Porcupine Bank. Your particle tracking figures, however, suggest near-slope patterns further south. Would velocity quiver maps on a shallower level maybe reveal any influx from the Bay of Biscay?*

As stated in our original reply (AC1), we decided to keep the quiver plots at 245m to conform with the acknowledged average core depth of the Slope Current, of approximately 200 – 300m (Porter et al, 2018). This choice has now been explained within the results section (lines 288 – 290).

*[original] Figs. 2-4:*

*You can state the association between T and S, and the tight linkage between T and SSH in the text, and only show the density field (Fig. 4). The T-S-density relations are well known between oceanographers, and the T and S figures are not strictly needed. And as suggested below, provide a better figure, which includes a relevant geographic domain, and averages over relevant periods.*

*I would also include altimetry data here. It would (i) validate the chosen in situ hydrography data product, (ii) produce and independent east ward transport record, and (iii) enable you to put your analysis in much better context – and link to the existing literature.*

*You lose out in insight and smear out valuable signal, by strictly averaging over these decades. For example, the 1990s was is contrasting period, with dense waters until the mid-1990s, and much warmer/lighter waters after. The average over these contrasting states is not so meaningful. I am aware of our wish to stay objective, but you have explainable reasons for selecting contrasting periods (e.g. early 1990s and early 2000s), in order to portray spatial hydrographic structures over the North Atlantic. Also pay attention to the (short term) interannual signal (mentioned above).*

We removed the original S (Fig. 2) and T (Fig. 3) plots as recommended. We have also extended the domain plotted to 80 °W - 0°W, 35 °N – 65 °N, which now better aligns with the plotted domain in the Ariane ensemble outputs (Figs 9 – 11). SSH data has been plotted in the same format (now subplotting as pre/post 1997, rather than the decadal mean anomalies).

*I would only show the GODAS-based Hovmöller diagram in Fig. 5. You say that there is mutual agreement between the GODAS-based and the EN4-based. I think there are large differences between them (although basic major feature are comparable). You also describe some limitations with the EN4 dataset (pages 10 and 11). And my impression of the hydrographic signal at ~60°N (which is based on many years of experience and many data sources), is that the GODAS product probably is more reliable for you purpose. Suggestion, skip EN4. It would enable you to produce a clearer figure, and convey a clearer message.*

EN4 has been removed from the manuscript. Whist both GODAS and EN4 have their limitations, we agree that GODAS is the more appropriate and reliable dataset.

*I would merge Figures 6 and 7 into one two-panel figure with the GODAS-based time series. It is reassuring that the EN4-based series show similar variability, and this could be mentioned with words/correlations.*

The original Figs. 6 and 7 have be merged to form a 2-panel plot of GODAS-derived eastward volume transport: a- geostrophic, b- total. This is presented in new Fig. 5.

*It is good to see the transport change in T-S space (Fig. 8). You could, however, zoom in on a narrower TS window, which would enable a better/clearer figure. The TS-transport figure based on ORCA12 (Fig. 9) is actually very different from the GODAS-based figure (Fig. 8). GODAS shows a major decrease around 5-6°C, 35.0 (which must be close to Subpolar Mode Water), which is not reproduce by the ORCA model. Suggestion: Stick to GODAS – skip Fig. 9.*

ORCA12 has been removed from this figure (now Fig. 7). The T and S ranges have been reduced to 34 – 36 for salinity, and 3.5 – 23 °C for temperature.

*Fig. 10. Yes the transports are much larger at the northern section (admixture of NAC-derived water, right?), and there is a somewhat worrisome decline in this transport (in line 255, you mention an almost-steady northward transport of 2 Sv after 1995, while I see a continuing decline, also after 1995). While I guess that you already have tested this thoroughly, are you still confident that you capture the entire slope current, with this model extraction? If yes, which current is then presently feeding the Faroe-Shetland Channel?*

The original Fig. 10 has been repurposed to better reflect the changes in northward transport along the shelf edge. Fig. 8 presents 4 northward transport time series: at 58, 57, 56, 53 °N. These now reflect the chosen Ariane shelf edge release locations, making them more relatable and useful in the wider context of our study.

The other comments from Hátún referred to minor amendments of figure titles and axis labels. The following comments were acted upon in full:

*Figure 1: Remove the header "Velocity quiver at 245 m" from each panel, and this common information in the caption. Remove the y-axis information on the right panels, and the x-axis information/labels in the upper panels, enlarge each panel, which removes too much void space between them.*

*Figure 5: Maybe use a bit narrower color ranges, in order to emphasize the obtained patterns.*

*The obtained Hovmöller diagrams based on GODAS and EN4 are actually rather dissimilar (mentioned above).*

*Figure 11-13*
*Just keep the y-axis information on the left panels, and the x-axis info on the upper panels. Enlarge the panels, which would remove the excessive white spaces in these figures.*

*Figure 6:*
*Remove "Geostrophic eastward transport", from the titles. This info is provided in the caption.*

*Figure 7:*
*Remove "total eastward transport", from the titles. This info is provided in the caption.*

*Figures 2-4: Remove " S/T/density decadal mean anomaly, and "205 m" from each title. This information is already in the caption.*

*Add some selected isobaths (e.g. 1000m, 2000 m and 3000 m) to these plots. In order to discuss these patterns against previously published key patterns in the North Atlantic (e.g. the spatial sea surface height mode, which is associated with the gyre index), you might what to include a broader meridional window (e.g. 35-65°N, although you only calculate transports over the 45-60°N latitudinal range). Maybe use a bit narrower color ranges, in order to emphasize the obtained patterns.*

For this final comment, we have been unable to add isobaths to the dataset due to time limitations for finding a suitable bathymetry dataset. We have extended the latitude range to plotted for 35 – 65 °N, and annotate the figures with the transect used in our calculation of transport (30 °W, 45-60°N).

**Our attention now turns to Reviewer 2's comments:**

*The ESC region is one of the best observed regions in the sub-polar North Atlantic. While data products such as EN4 and GODAS provide a full 4D overview of the region of interest. They are also often not great. The authors acknowledge this to some degree, although I find the statements on this quite confusing. In Section 2.1, the authors highlight that GODAS salinity is mostly "synthetic" and "seriously under estimates salinity variability", but in the discussion in Section 4.1, the EN4 lack of salinity data and gridding methods is flagged as a potential issue. I also find the assumption in lines 107-111 requires further evidence that it is appropriate.*
*Line 114-115: The two data products are stated to be independent of each other, but I doubt this is truly the case (e.g. if both incorporate Argo profiles). Particularly the following sentence highlights that these are likely the same four sources (please state here which ones also!).*

Combining this and Hátún's comments as already discussed, we removed EN4 from our manuscript. We have tried to better explain the limitations of GODAS in methods section 2.1, and the methods used to collect and assimilate GODAS data. We have provided further justification on why GODAS is an appropriate dataset to use in this region; for the purposes of assessing the hydrography of the North Atlantic basin and estimating geostrophic volume transport estimates towards the shelf edge.

*The paper is highly descriptive of what is going on, but lacks to place this into the context of the forcing mechanisms. For example, there is no consideration for the positioning of the sub-polar front in the North Atlantic, there is also no consideration for the wind-forcing of the circulation of the wider SPNA and how this influences "recruitment" into the ESC or otherwise. Especially given the discussion on zonal current variability, I find these quite major omissions in the analysis. The discussion is more a continued description of the results presented, rather than any contextualisation in terms of previous work and/or forcing mechanisms. Section 4.2 is more speculative on implications, and a repeat of what has already been stated in the introductions.*

We now discuss our results in context with previous published works on subpolar North Atlantic forcing: SSH dynamics, buoyancy forcing, the effects of the NAO on wind forcing and the gyre index (section 4.1). We also look at how changing wind stress in the vicinity of the European shelf break forces changes in the Slope Current via Ekman transport towards the shelf break. We can conclude that the changes in Atlantic inflow and the Slope Current are not associated with direct effects of wind forcing, either local or basin-scale, but are rather driven by slower changes in density (see lines 271-279, 386-387).

*The authors rely on the data products to provide accurate baroclinic transport, but based on potentially erroneous salinity data. There is little quantification of salinity error or overall error*

*analysis, it is therefore difficult to know whether this really is of no significance to the results presented.*

We have further emphasized that density variability is dominated by temperature variability in the subpolar North Atlantic. Repeating the thermal wind analysis with climatological salinity (not shown in our analysis), we can evidence that uncertainty in the GODAS salinity data does not substantially affect our results or conclusions (methods section 2.1).

*The paper lacks visual cues of the lines/boxes etc used, as well as a figure that highlights of the focus area of the study sits within the Sub-Polar North Atlantic (SPNA). Even for someone with expertise in the region, it is difficult at times to follow which transect has been used or across which box particles have been quantified. None of the figures show the "analysis region" (line 165) in full, for example.*

We have extended the geographical domain plotted on some figures (for example: Figs 2 and 3) so they show the entire region of interest. On both of these, we have annotated the 30 °W, 45 – 60 °N eastward transport section. Figure 1 is also annotated with the four shelf edge northward transport profiles, for which we show transports in Fig. 8.

*Line 34—36: Johnson and colleagues find that the changes in the water mass properties and nutrients concentrations at the Extended Ellett Line are related to changes in properties of the circulation. To my mind, this is not the same as changes in concentrations in upstream flows, as the authors state.*

We have emphasized that properties and flows are linked, as evidenced in our analysis by a weaker, warmer Slope Current in the early 2000s.

*Line 46-47: Suggest rewrite for clarity "The Gulf Stream flows between these two and eventually ..."
Line 51: "However, not all of the water follows this pathway." (missing "of").*

These minor issues have been fixed.

*Line 52-56: This description neglects some of the other exchanges in the northern North Sea, particularly the Norwegian Trench inflow. The authors have spent great length emphasising the importance of the ESC to the marine ecosystem of the continental shelf, so a correct description here is warranted.*

The Norwegian Trench Inflow has now been added in lines 55-57.

*Line 67-69: This reduction in temperature was also accompanied with some very strong reductions in salinity. The region of the ESC was at its freshest for more than 120 years.
Please see Holliday et al., 2020.*

We have highlighted declining salinity in the ESC and wider SPG region due to an extreme negative NAO state, referencing this paper.

*Further in the paper, there is discussion of the baroclinic and barotropic components of transport. I think this could be clarified here in the methodology.
Line 135: Why not state x 10-6. The e-notation seems a relic of the coding.*

The methods section has been updated with a new section 2.2 "transport calculations and metrics", with further details on the transport components.

*Line 150-167: I found the description of the particle tracking methodology could be better: it is very detailed about some things (e.g. reference the initial positions file), but lacked details on other. Are particles released from all grid cells? Or only grid cells following the continental shelf edge? Later in the paper there is also mention of particles crossing certain transects. I would recommend some major rewrite of this section to ensure transparency and repeatability.*

Section 2.4 has been re-written to provide a more detailed account of our Ariane particle tracking experiments, addressing the points above.

*Line 231-247 (and probably throughout): Practical Salinity is a unitless quantity. It should be used as such. Therefore text should say "Practical salinity in the range 34.25-36 …").*
*That being said, oceanographers agreed to adopt the TEOS-10 convention, and the authors already use the Gibbs Python functions, so Absolute Salinity should be used.*

We confirm that salinity as provided by GODAS is in units kg/kg, akin to absolute salinity (units g/kg); we have clarified this in the revised manuscript and no longer refer to "PSU".

*Line 250-252: Recommend to plot these transect locations on a map. If not on a general overview map, at least on one of the pre-ceding figures.*

As previously stated, Figs. 1 – 3 now have the relevant transport transects annotated.

*Line 250-288 (Section 3.3): This section is very descriptive, but lacks interpretation (here or later in the discussion) on how this relates to what is already known of the region's circulation. I was unclear what the authors consider the novel finding from this analysis.*

We have tried to provide a more quantitative perspective and to emphasize our key finding that the inflows feeding the Slope Current are systematically different when we consider years of 'cold/strong' and 'warm/weak' Slope Current transport throughout the results and discussion. We highlight that the density driven currents are the main source of shelf edge flow variability.

*Figure 1: A continuous colour bar is not helpful to the reader. I would suggest using fewer, more discrete intervals in the colour scheme. The quiver is also quite difficult to see and may need to be scaled up. Which months do the authors consider winter/summer? [I note the colour bar does improve in future figures]*

*Figure 2/3: The decadal distinctions are a human reflection of the calendar, rather than a reflection of the physical ocean climate. The "warm/cold phases" are not specifically associated with the changes from the 90s to the 00s – for example with major changes happening mid-1990s. The authors should consider using a more objective way of combing years into more meaningful "warm/cold" or "strong/weak" composites.*

Original Figs 2-4 have been amended (as previously described) and now show the shift between the warm and cool periods of pre and post 1997 respectively. This better aligns with the T-S figure presented, and also better link back to the observed temperature shift in the previous literature and our own study.

We hope that the revisions we have made in response to the reviewer comments, as detailed above, act to satisfy both reviewers. We look forward to hearing the comments in the next review period.

Matt Clark, Lead Author
On behalf of the authors.

---

## Author Response (AR2)

**Author's response to reviewer comments, from the second review**

Many thanks to Hjálmar Hátún and an anonymous second reviewer for their constructive comments on our manuscript. We will begin by responding to Hátún's annotated PDF of our manuscript, before moving onto the comments provided by Reviewer 2. Original reviewer comments are provided in *grey italics*.

The authors have responded to my comments/concerns in satisfactory ways. A few additional comments are provided as annotations to the attached pdf.

We were happy to hear that Hátún was satisfied by our previous responses, and grateful for the provided further annotations on our manuscript. The annotations highlighted some minor points which we had overlooked, such as minor typos, formatting issues with symbols used in the figures (1, 2, 4, 6, 7) and equations, and incorrect (or incomplete) references. These have all been amended as recommended. In response to the comment:

I guess the series stop in 2010, since this is when the model run terminated, right? It is a pity that that we do not get the change to see if the transport might have intensified after 2015. I, however, understand that there will not be time to run this model up to date.

This is correct. Sadly due to computer and data access issues with the JASMIN computer system we had earlier in the analysis, we were only able to obtain the model run up to 2010.

Moving onto Reviewer 2's comments, we were disappointed that the reviewer found our previous responses unsatisfactory.

**1) GODAS Description**

In my opinion, it is incorrect to call GODAS an "observational dataset" (section title, line 116). It is an ocean reanalysis product, and should be termed as such. Apologies for not noticing in the initial submission. My not noticing in the initial submission led me to make some comments which were not necessary. However, I believe it's critical to acknowledge ocean reanalysis correctly. Line 111-112 needs revision to reflect that they are using "an ocean reanalysis product and an eddy-resolving global ocean model hindcast".

**The reviewer is correct in saying that GODAS is an ocean reanalysis product. We have now amended the manuscript to reflect this, primarily in the methods section.**

At the start of this description, the authors refer to "measurements of…" This is not strictly true, as the measurements have been assimilated into a numerical model (the GFDL Modular Ocean Model). Therefore the output of the reanalysis is not a measurement. I suggest replacing with "values of …" when referring to the GODAS outputs (sentence line 118-119). On line 119-121, the correct verb in this sentence should be "assimilates". Line 126-127: GODAS does not interpolate, it is a reanalysis product, using a data assimilation scheme (according to the documentation a 3DVAR one), it is not an interpolation scheme.

Throughout the manuscript, we have now stated that GODAS data are "values" instead of "measurements". Line 119 has been amended to state "assimilates". In the methods section 2.1, we further clarify that GODAS assimilated measurements using 3DVAR assimilation.

On lines 183-184, the authors refer to GODAS "altimetry data". This is not strictly correct, while GODAS assimilates the satellite altimetry data after 2007, this is not altimetry data. Even after 2007, the GODAS SSH field will be a numerical solution that balances all input data in the data assimilation scheme. The GODAS product refers to a "sea surface height relative to the geoid", and the description should refer to this.

**The reviewer is correct and we apologise for not spotting this in our initial submissions. We have now changed the description on line 179 to now read "GODAS sea surface height relative to the geoid (SSHG)".**

Finally, further suggested changes relating to this comment are: line 186-187, line 384-386, line 423; where the authors refer to "observational data". On line 409, the authors use "observations" to mean "results". I suggest replacing; "observations" as it is a loaded word in our subject area (I hope my comments above sufficiently illustrate), particularly where comparisons with numerical model outputs are made.

**Where appropriate we now refer to "values" rather than "observations".**

**2) NCEP wind fields.**

The authors have now also applied a new data source, wind fields from NCEP. It is unclear whether they use this to refer to wind fields which may be included as part of the GODAS data distribution, or whether they have sourced this elsewhere. Suitable reference/acknowledgement to the underlying data and its provider should be made, as well as a short description (a meteorological reanalysis, or an observational dataset?).

**In Section 2.2 lines 160 - 163, we now introduce the NCEP wind with a description of the grid resolution and data source.**

**3) Lines 146-167 / Transport calculation**

The authors have much improved this section, however, I still found it implicit as to how they have calculated the total transport. Based on the description and equations (1) and (2), it is implied that the total eastward volume transport is the integral in equation (2) through the section from the eastward velocities from the GODAS reanalysis product, and the geostrophic eastward volume transport is based on thermal wind. It was my own error in not realising the reanalysis product is a model which also provides the absolute velocities, and therefore my comments in the initial review referring to no mention of the geostrophic referencing were due to not realising this.

We have now further amended the manuscript to clarify the differences between total and geostrophic transport. Equation 2 has been updated to reflect that this is the eastward geostrophic transport equation, and lines 158 – 159 reflect how we use the GODAS eastward component of currents to calculate the total absolute transport through the same region.

**4) The choice of sections for calculation of transports**

The choices of sections are at times difficult to interpret based in knowledge from the circulation of the region. At 30 W, the authors define transports between 45 N and 60 N, over 0 to 1000 m. This will include some of the recirculating branches in the sub-polar North Atlantic, and will include some transport which does not contribute to the ECS. Based on the histograms in Figures 12 & 13, 55 N as the northern boundary would have been a more considered choice in terms of how circulation at 30W feeds the ECS transport.

It is also not clear to me why the authors decided to define the slope current transport as solely the northward transport component. In my opinion, it make it less meaningful to compare to established calculations of slope current transport in the published literature (Porter, Hopkins, Huthnance and others).

We have shown in our results that the transports seen at our 30 °W 45 – 60 °N meridional transects show strong similarity in terms of changes to the transport over time in our northward transects at the shelf edge (Figure 8). Of course, not all the transport in this region will reach the shelf edge.

**5) Other minor comments**

Lines 196-237. The description of particle tracking is much improved. However, it would still be helpful to have a range of the number of particles released in each experiment.

In the methods section 2.4, lines 206 - 208, we have now stated the number of particles released in our experiments. We have also highlighted that the large difference in particle releases is directly related to the reduction of transport observed in our results.

Lines 248-249: I am presuming TEOS10, but this choice of words leaves that open for interpretation. I suggest omitting "... using the equation of state of seawater" and referring back to the methodology "(Section 2.2)"

Our analysis does indeed use TEOS10. We have referenced back to Section 2.2, and mentioned we are using TEOS10.

Other typos mentioned have been amended. The following figures were amended in response to:

Figure 1. Previous comments relating to this figure have not been adequately addressed. The image quality in the PDF file was still poor. I would suggest improving the scaling of the quiver to make arrows more obvious. There is no indication on months used for summer/winter. The caption states "relative velocity" but not what to; the text states the figure shows "quiver plots of velocity".

Figure 8. Caption states that transects are both ~100 km (first sentence) and 50 km (second sentence) long. Please clarify.

Figure 1 has now got arrows that are thicker and scaled by 2. The caption now states that summer refers to July, and winter refers to January.

The caption in Figure 8 has been amended to state the length of each zonal transect.

We hope that these amendments satisfy both reviewers and the editor. We look forward to hearing back.

Matt Clark, on behalf of all co-authors.

---

## Author Response (AR3)

**Author's response to manuscript acceptance**

Many thanks to Matthew Hecht for accepting our manuscript for publication.

As requested, we have amended Figure 1 to make it clearer. The quivers have been made bolder and the range of the plot axis has been reduced to focus directly on shelf-edge flow.

An "author contributions" section has now been added to comply with the Copernicus publishing guidance. No other amendments have been made since the last review.

Kind regards,

Matt Clark, on behalf of all co-authors.